# A de novo strategy for predictive crystal engineering to tune excitonic coupling

Ritesh Haldar [1], Antoine Mazel[2], Marjan Krstić [3], Qiang Zhang[1,3], Marius Jakoby[4], Ian A. Howard [4,5], Bryce S. Richards[4,5], Nicole Jung[6], Denis Jacquemin [2], Stéphane Diring[2], Wolfgang Wenzel[3], Fabrice Odobel[2] & Christof Wöll [1]

In molecular solids, the intense photoluminescence (PL) observed for solvated dye molecules is often suppressed by nonradiative decay processes introduced by excitonic coupling to adjacent chromophores. We have developed a strategy to avoid this undesirable PL quenching by optimizing the chromophore packing. We integrated the photoactive compounds into metal-organic frameworks (MOFs) and tuned the molecular alignment by introducing adjustable "steric control units" (SCUs). We determined the optimal alignment of core-substituted naphthalenediimides (cNDIs) to yield highly emissive J-aggregates by a computational analysis. Then, we created a large library of handle-equipped MOF chromophoric linkers and computationally screened for the best SCUs. A thorough photophysical characterization confirmed the formation of J-aggregates with bright green emission, with unprecedented photoluminescent quantum yields for crystalline NDI-based materials. This data demonstrates the viability of MOF-based crystal engineering approaches that can be universally applied to tailor the photophysical properties of organic semiconductor materials.

---

[1] Karlsruhe Institute of Technology (KIT), Institute of Functional Interfaces (IFG), Hermann-von-Helmholtz Platz-1, 76344 Eggenstein-Leopoldshafen, Germany. [2] Université de Nantes, CNRS, Chimie et Interdisciplinarité: Synthèse, Analyse, Modélisation (CEISAM), UMR 6230, 2 rue de la H oussinière, 44322 Nantes Cedex 3, France. [3] Karlsruhe Institute of Technology (KIT), Institute of Nanotechnology (INT), Karlsruhe Institute of Technology (KIT), 76344 Eggenstein-Leopoldshafen, Germany. [4] Karlsruhe Institute of Technology (KIT), Institute of Microstructure Technology (IMT), Hermann-von-Helmholtz Platz-1, 76344 Eggenstein-Leopoldshafen, Germany. [5] Karlsruhe Institute of Technology (KIT), Light Technology Institute (LTI), Engesserstrasse 13, 76131 Karlsruhe, Germany. [6] Karlsruhe Institute of Technology (KIT), Institute of Organic Chemistry (IOC), Fritz-Haber-Weg 6, 76131 Karlsruhe, Germany. Correspondence and requests for materials should be addressed to S.D. (email: stephane.diring@univ-nantes.fr) or to W.W. (email: wolfgang.wenzel@kit.edu) or to F.O. (email: fabrice.odobel@univ-nantes.fr) or to C.W. (email: christof.woell@kit.edu)

Many applications[1–3] require the development of highly emissive chromophoric assemblies, but self-assembly of fluorescent chromophores into mesoscale-ordered structures often encounters problems resulting from the presence of dominant non-radiative decay paths leading to photoluminescence (PL) quenching[4,5]. A typical case is the cofacial π–π stacking, a frequent structural motif in the crystal structure of planar aromatic molecules. In these aggregates, the absorbed excitation energy is transferred to the lowest excited state in the neighboring chromophore. In many cases, also referred to as H-type non-emissive aggregates[5], fluorescence is prohibited by dipole selection rules, and the excitation decays via non-radiative processes. Recovering the fluorescence present for the isolated monomers is of pronounced interest. J-type aggregates containing slipped or head-to-tail arranged π–π stacks of chromophores are highly fluorescent and of substantial interest as organic light-emitting materials[6–8]. As J-aggregates also feature high photo-absorption efficiency and reduced charge-recombination losses, they are also of pronounced interest for organic photovoltaics[9,10]. This general interest has stimulated a large number of efforts to enforce J-type aggregation of chromophores in supramolecular assemblies[11–13]. However, very often the resulting assemblies do not have the desired properties, instead of high luminescence, non-radiative quenching processes result in "dark" materials[4]. Apart from the H-aggregation, presence of transition metal-ions can also quench the emission[14]. Many of these approaches were realized by modifying intermolecular interactions via the introduction of H-bonding side groups or moieties, which affect the molecular packing through steric requirements[6–8]. The difficulties of crystal structure prediction (CSP)[15,16], however, have severely limited the application of rational strategies and reported J-aggregates are mostly the results of serendipitous discovery[6–8,11–13,17]. A typical example is the case of 9-anthracene carboxylic acid (ACA), a small, prototype chromophore. Non-substituted ACA crystallizes in a triclinic structure, with bright PL properties. When side groups are attached at the 10-position of ACA to modify the position of emission bands, the slightly modified chromophores crystallize in rather different crystal structures (triclinic, monoclinic, and orthorhombic)[18]. As the CSP method struggle to predict experimentally realized structures, researchers had to resort to trial-and-error strategies to optimize the photophysical properties of such ACA assemblies.

Here, we propose a rational, de novo strategy for tuning chromophore packing that is based on metal-organic frameworks (MOFs). MOFs are constituted by coordinating organic linkers (in this case chromophores with suitable coordination groups) to metal or metal/oxo nodes, yielding porous, highly regular structures[19–21]. In these crystalline coordination networks, the structural freedom of the chromophore packing is highly reduced, since the overall topology of the MOF mainly depends on the connectivity of nodes and linkers. The porous structure of MOFs provides sufficient free space to allow for rotations and other conformational changes of the linkers within the bulk structure[22,23]. Such changes can be induced by attaching functional side groups to the chromophoric linkers. Since these modifications do not affect the topology of the assembly[24,25], CSP can be carried out in a phase space of reduced dimensionality, making it much more accurate than for molecular crystals[26,27].

Herein, we demonstrate our de novo approach for predictive tuning of structural parameters in a crystalline chromophore assembly for a particular MOF, Zn-SURMOF-2 [28,29]. A Zn-based paddle-wheel-type secondary building unit is tethered to a ditopic, chromophore linker, yielding a square grid type two-dimensional structure of closely packed sheets (Fig. 1a)[28–30]. In the present study, we have focused on a unique class of chromophores, naphthalenediimides (NDIs). In addition to

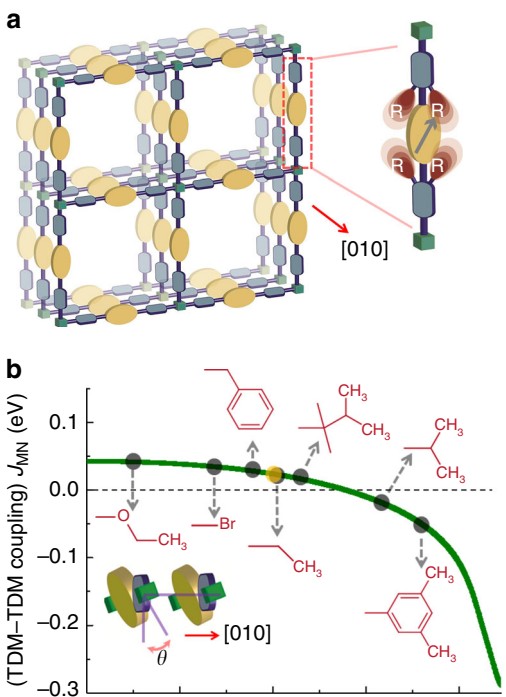

**Fig. 1** Prediction of Coulomb coupling in SURMOF-2 structure. **a** Schematic of a Zn-SURMOF-2 structure showing the continuously stacked chromophores along the [010] plane, with green box = Zn-paddle-wheel type secondary building unit, yellow ellipsoid = NDI(OEt)$_2$, black arrow = transition dipole orientation. The schematic at the right-hand side illustrates the linker with the"SCU" R. **b** The green line is a plot of calculated (transition charge from electrostatic potential method, TrEsp) Coulomb coupling vs rotation angle θ (as shown in the inset). The different R groups that exert different values of θ are illustrated on the plotted graph. The yellow sphere marked on the green line is the predicted θ (by GROMACS-2018.4) without any SCU (R = H); The inset figure shows the rotation angle θ controlled by the R groups

unsurpassed PL quantum yields (QYs) in solution, these organic dyes are able to form of charged-transfer complexes and exhibit high charge carrier mobilities as well as other interesting photophysical properties[31–33]. The photophysical properties of NDI monomers can be rationally tuned by substituting the core with electron-pushing or -pulling groups[34]. Despite their huge potential, so far only a few supramolecular assemblies of such core-substituted NDIs (cNDIs) have been explored[35–37], since the design of efficient cNDI-based chromophoric assemblies is limited by tedious synthesis procedures and unpredictable self-assembly[35–39].

As a prototype linker for constructing chromophoric MOFs, here we use a bis-ethoxy-substituted-NDI (NDI(OEt)$_2$), equipped with two carboxylic acids. This linker can be readily assembled into a SURMOF-2 structure, where π–π interactions yield a rather close packing (Fig. 1a). As expected in the case of such a close packing of chromophores, non-radiative processes govern the decay of optical excitations, and the corresponding SURMOF-2 is a non-emissive H-aggregate[40]. When we inspected the structure in more detail, we realized that H-aggregate formation could be suppressed by increasing the rotation angle θ of the chromophoric linkers around the molecular axis (Fig. 1b). Our heuristic expectation that larger distances between NDI core planes and a slipping of the intermolecular transition dipoles, resulting from

larger values of $\theta$, should lead to a reduction in fluorescence quenching was confirmed by a theoretical analysis, see Fig. 1b. These calculations (for details see the Methods section) were based on the transition charge fit (TrEsp) and revealed that for $\theta > 55.4°$ the Coulomb coupling changed sign, indicating a transition from H- to J-type aggregation. Note that we did not consider charge-transfer coupling, because the inter-cNDI distance is much larger than 4 Å (required for a tight molecular packing).

In the present work, to tune $\theta$ to the desired value, we implement an engineering approach by attaching "steric control units" (SCUs) to the imide -N of the cNDI, as shown in Fig. 1b. Unfortunately, the tedious synthesis of cNDI monomers prohibits a trial-and-error approach to investigate the effect of these SCUs. Therefore, we first create a library of 18 possible SCUs, then optimize the geometry of the individual linkers using a force-field calculation, and then use a simple scheme to assemble 18 different MOFs in silico using a previously described MOF constructor[27]. Then, the MOF lattice constant is fixed and an MD scheme is used to optimize the structure of the linkers (including the intramolecular dihedral angles). As a result of inter-ligand interactions, the dihedral angles are changed and also the rotation angle $\theta$ (angle between cNDI core and carboxylate-planes) changed. The corresponding results (Fig. 1b) led to some surprises, e.g. the fairly bulky benzyl group exerted smaller dihedral and rotation angle compared to a $-CH_3$ substituent. This computational screening process identifies three suitable, synthetically viable SCUs, which are then synthesized and assembled into the Zn-SURMOF-2 structures displayed in Fig. 1a. The synthesized Zn-SURMOF-2 displays J-aggregation feature with bright PL, as predicted by computational methods.

## Results

**Structural prediction of a library of NDI(OEt)$_2$-assemblies.** To arrange the cNDIs in space, we focused on MOFs of type Zn-SURMOF-2, assembled from ditopic (with phenylcarboxylate coupling groups) cNDI linkers (see Fig. 1a)[28,29]. A few cNDI-based bulk MOF structures have been reported previously, but either their photophysical properties were not studied[41,42] or only non-emissive H-aggregates were obtained[40].

To study the interplay between chromophore packing and photophysical properties, we started with a NDI(OEt)$_2$ linker (Fig. 2a and Supplementary Fig. 1). Using the DFT method we have optimized a trimer model of NDI(OEt)$_2$ (**A**). These calculations yielded a cofacial stacking ($\sim$5.8 Å) of the NDI(OEt)$_2$ cores, an undesirable H-aggregate type arrangement (see Methods section and Supplementary Fig. 2). The mounting of the NDI(OEt)$_2$-linker inside the framework (via the two phenylcarboxylate groups) offers the option for tuning the inter-NDI(OEt)$_2$ coupling by changing the rotation angle $\theta$ (see Fig. 1b). To illustrate this effect, we employed the electrostatic potential (TrEsp) method[43]. The TDM–TDM (transition dipole moment) Coulomb coupling between two closely spaced ($\sim$6.8 Å) NDI(OEt)$_2$ chromophores was calculated as a function of rotation angle $\theta$. Since such calculations are not possible for a periodic structure, the situation inside the MOF was approximated by studying two adjacent NDI(OEt)$_2$ monomers placed at a distance ($x$) of 6.8 Å between the molecular axis. While for $\theta \sim 2°$ the coupling in these dimers was positive, explaining the H-aggregate type behavior in A (Supplementary Table 2), for $\theta > 55.4°$ the interchromophore coupling sign reversed, indicating the formation of a J-aggregate. For larger distances ($x$), the coupling strength decreased, but the nature of coupling (H or J) remained the same (Supplementary Fig. 3).

Next, we attached sterically demanding side groups R to the cNDI core, as a synthetic strategy to tune $\theta$. Depending on their

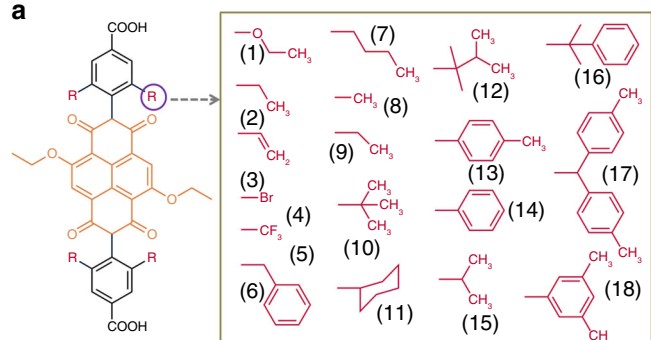

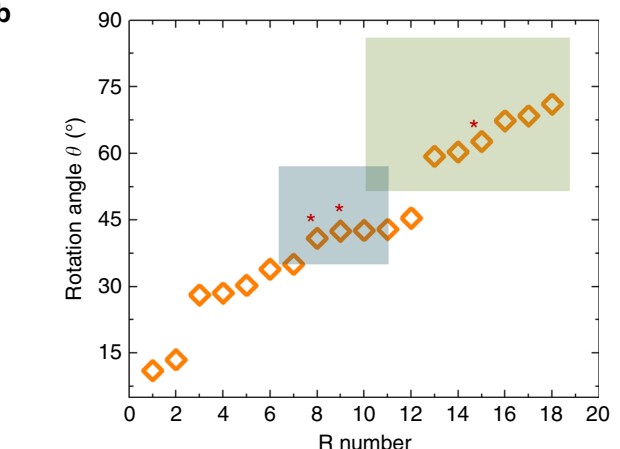

**Fig. 2** Design of "steric control unit". **a** Chemical structure of R-NDI(OEt)$_2$ and the possible R groups. **b** Plot of the rotation angle $\theta$ (calculated using GROMACS-2018.4 package, using a trimer model) vs the different R groups numbered as in a. The blue region is the borderline, and the green region corresponds to the possible J-aggregates. *Marked groups were synthesized and assembled as SURMOF structures

precise geometry (Fig. 1b), the R moieties act as SCUs by changing the dihedral angle ($\alpha 1$) between the NDI(OEt)$_2$ plane and the phenyl plane, and thus affect the NDI core rotation angle $\theta$. We created a library of 18 possible R groups (chosen such that they leave the optical properties of the chromophore unchanged) and calculated the resulting values of $\theta$ (Fig. 2a, b and Supplementary Table 1). These simulations (using a trimer model) revealed that for these R-groups rotation angles between 0° and the maximum limit of $\sim$90° can be achieved (Fig. 2b). It is worth noting that the dihedral angle ($\alpha 1$) and rotation angle $\theta$ show a nontrivial dependence on the "bulkiness" of R, where e.g., the methyl (8) group exerted larger dihedral and rotation angle (89.7° and 40.9°, respectively) as by the benzyl (6) group (87.6° and 34°, respectively) (Fig. 2b and Supplementary Table 1).

To validate the predictions for chromophore geometry and excitonic coupling using the DFT method, we chose three R groups, two from the borderline region in Fig. 2b (8, R = Me and 9, R = Et from the blue region), and one for which a J-aggregate is predicted (15, R = iPr from the green region) (Fig. 2b), for subsequent experimental analysis.

**Crystalline assembly of R-NDI(OEt)$_2$ in Zn-SURMOF-2 structure.** The three linkers obtained from the screening process described above were synthesized according to the established cNDI synthetic protocol[40]. Subsequently, using a layer-by-layer (lbl) spin-coating method, we fabricated Zn-(R-NDI(OEt$_2$))

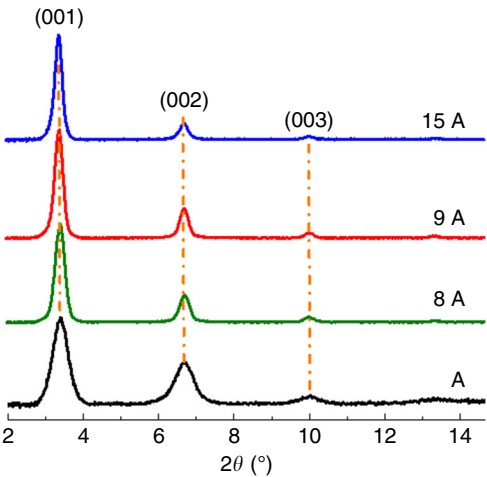

**Fig. 3** Assembly of R-NDI(OEt)$_2$ in SURMOF-2 structures: out-of-plane XRD patterns of A, 8A, 9A, and 15A

(R = Me (8 A), Et (9 A), and iPr (15 A)) SURMOF-2 structures on –OH functionalized quartz and silicon substrates (see Methods section). The out-of-plane and in-plane X-ray diffraction (XRD) patterns of 8A revealed the successful growth of crystalline MOF thin films that were oriented along the (001) axis, with lattice dimensions ($a = b = 2.6$ nm) perpendicular and parallel to the substrate that were identical to those of the parent Zn-(NDI(OEt$_2$)) SURMOF-2 structure (A) described previously (Fig. 3 and Supplementary Fig. 4)[40]. The other two SURMOF-2 structures, 9A and 15A, also revealed XRD patterns that were similar to that of 8A, suggesting identical structures. In the case of 15A, which contained the most bulky SCU –iPr, the interlayer distance was found to increase to ~6.8 Å, 1 Å greater than that in A (~5.8 Å) (Supplementary Fig. 4). The interlayer distances for 8A and 9A were also similar (6.6–6.7 Å). This interlayer distance, however, did not affect the aggregation type (coupling sign), as evidenced by calculations of the Coulomb coupling for different distances. The DFT simulated linker orientation angles are different (as expected) from the GROMACS-optimized values, see Supplementary Table 2. Supplementary Figure 5 shows the possible geometry of the NDI(OEt)$_2$ linkers in structure A and 15A (by considering a model of trimer), illustrating the changes in dihedral angles caused by the SCU.

**Effect of "steric control units" on excitonic coupling**. The UV–Vis absorption spectra of all the linkers solvated in ethanol were similar, confirming that the "steric control" R-groups do not affect the optical properties of NDI(OEt$_2$) (Supplementary Fig. 6)[44]. However, after assembling the modified linkers into the corresponding SURMOF-2 structures, the absorption spectra exhibited pronounced differences, indicating substantial differences in the excitonic coupling (Fig. 4a). First, the ratios of the vibronic overtones at 436 ($A_{(0-1)}$) and 472 ($A_{(0-0)}$) nm (Fig. 4a) were strongly different. In the case of 8A, the vibronic ratio resembles that in A, but an additional blueshift was observed. Interestingly, for 9A and 15A, an inversion of vibronic intensity was observed. Considering the vibronic features observed for typical Kasha aggregates[5,8], the stronger intensity of the first overtones ($A_{(0-0)}$) in 9A and 15A clearly suggest loss of cofacial stacking of the NDI(OEt)$_2$ chromophores[6]. We noted that the coupling energy ($J_{mn}$) for A (+22 meV) and 9A (+21 meV) or 15A (−16 meV) did not differ markedly (Fig. 1b), and hence a large spectral shift would not be anticipated. Rather the

changes observed in the vibronic ratios indicated a different aggregation behavior. Particularly, the absorption spectrum of 15A shows a broader peak in the longer wavelength region (>500 nm) compared to the other SURMOFs. To explain this behavior we employed time-dependent density functional theory (TD-DFT) to simulate the electronic absorption spectra of the NDI(OEt)$_2$ dimers with rotation angles 34° and 60° for H- and J-type coupling, respectively (Supplementary Figs. 7 and 8). In the case of H-type, the absorption maximum shifted slightly to a shorter wavelength. But, in J-type a new absorption shoulder was observed in the longer wavelength region (~485 nm). This is because in a J-aggregate two transition dipoles become almost collinear, and the dark state (in H-aggregate) becomes allowed and low lying visible absorption peaks are observed in the spectrum. These data agreed well with the experimentally observed absorption spectrum of 15A.

The differences in the PL spectra upon excitation at 450 nm were even more dramatic. In contrast to the non-emissive structure A, 8A and 9A exhibited substantial PL intensities, with a maximum at 480 nm and a broad band at longer wavelength (520–600 nm) (Fig. 4b). In the case of 15A (blue), PL became much more intense; also the 480 nm peak became less prominent and was overtaken by a broad PL peak with a maximum at 540 nm. The PL maximum at 480 nm resembled that observed for non-coordinated linkers in solution. We attributed the red shifted PL in these SURMOFs to the formation of J-aggregates. We can rule out the possibility of excited state dimer (excimer)[45,46] formation: the absorption vibronic ratios do not resemble that of the solvated chromophore and the excitation spectrum monitored at 580 nm exhibited a broader peak ~500 nm compared to that of the solvated chromophore (Supplementary Fig. 9). This outcome agrees well with our prediction, as the "steric control" 15 rendered J-type coupling, and the observed intense red shifted PL supports the hypothesis.

The PL lifetimes recorded for 8A, 9A, and 15A exhibited two components with different lifetimes of ~6.5 and ~1.3 ns (Fig. 4c). Whereas in 8A and 9A the short-lived component dominated the emission, in 15 A, the long-lived PL[47] became the more prominent component. This observation is consistent with the larger rotation angle $\theta$, which would lead to a loss of the cofacial packing. Photographic images of the linkers in solution and of the corresponding SURMOFs are given in supplementary Fig. 10 (see also Supplementary Fig. 11). Whereas the bright cyan emission of the solvated linker PL remains unaffected by attaching the "steric control" group, the SURMOFs reveal striking changes. Attachment of the SCUs transformed the non-emissive A to an intensely green emissive thin film 15A. The PL QY measured for 8A, 9A, and 15A (~0.4%, 0.7%, and 2.3%, respectively) was consistent with the formation of J-type assemblies (Fig. 4d).

## Discussion

Our results reveal that a MOF-based approach using chromophoric linkers allows for a rational crystal engineering of highly regular chromophoric assemblies. Starting from a non-emissive H-aggregate, where the close proximity of neighboring chromophores strongly quenches optical excitations, we have used computational methods to search for SCUs to modify the linker packing, with the aim to realize J-aggregates with bright emission. Using a library of chromophoric MOF linkers differing in their SCUs, we were able to identify functionalities, which change the molecular packing such that the π–π cofacial stacking is strongly reduced. This search was only possible because in this MOF-based approach prediction of the linker (chromophore) arrangement is largely simplified by the fixed framework topology, where effectively only one degree of freedom remains. The

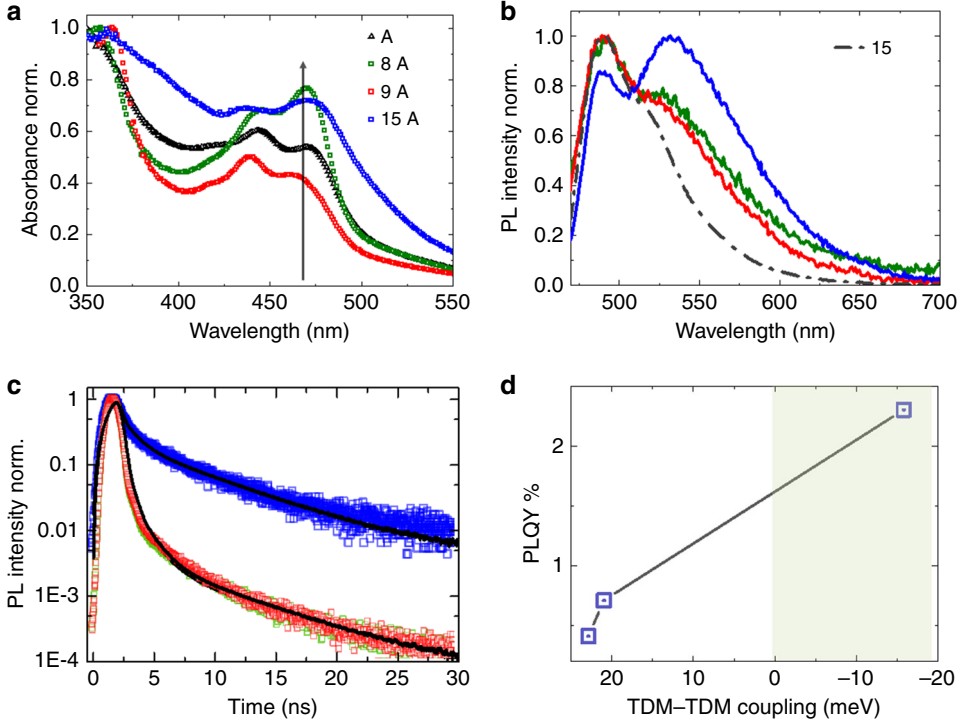

**Fig. 4** Photophysical properties of the SURMOFs. **a** Absorption spectra of 8A, 9A, and 15A illustrating the different vibronic ratios compared to that of the A, the black arrow indicates the position of the $A_{(0-0)}$ transition in solvated $NDI(OEt)_2$ chromophores irrespective of the different R substituents. **b** PL spectra of 8A, 9A, and 15A and iPr-$NDI(OEt)_2$ (15) in ethanol (20 µM) upon excitation at 450 nm, recorded at room temperature, with the color code being the same as in panel **a**. **c** PL decay of 8A, 9A, and 15A. Color code is same as in panel **a**, and the black lines correspond to the exponential fits. **d** Plot of experimentally obtained PLQY vs predicted TDM–TDM coupling (as in Fig. 1b); the green marked area indicates the J-type coupling

computational screening procedure reduced the dipole–dipole coupling of the non-emissive start structure (~+22 meV) to −16 meV.

The theoretical prediction was fully confirmed by a thorough photophysical characterization of the SURMOFs built from the modified organic linkers identified in the computational screening. The optical characterization of the target material was greatly aided by the structural properties of SURMOFs as compared to systems made from MOF powders[28,29]. In the latter case, the morphological heterogeneity and the irregular nanoscale morphology severely hamper the optical and photophysical characterization. The non-emissive starting material was converted into a highly emissive SURMOF with bright green luminescence. The possibility to rationally design J-aggregates from starting chromophores is of tremendous interest from the perspective of exciton transport, electron transport, and other related optoelectronic properties.

## Methods

**X-ray diffraction**. The XRD measurements for out-of-plane (co-planar orientation) were carried out using a Bruker D8-Advance diffractometer equipped with a position-sensitive detector LynxEye in geometry, operated with a variable divergence slit and a 2.3° Soller-slit on the secondary side. Cu $K\alpha_{1,2}$-radiation ($\lambda$ = 0.154018 nm) was used in all cases.

**Computational methods**. The Coulomb couplings (TDM–TDM coupling shown in Fig. 1b as a green line) were calculated using transition charge from electrostatic potential (TrEsp) fit (where transition charges were fitted to every atom in the monomer from electrostatic potential fit)[43]. First we calculated transition density matrix between ground state $S_0$ and excited state $S_1$ and save it in "cube" format. From this point we employed MultiWfn program[48] to make the fit and calculate transition charges for every atom in monomer. These transition charges have been used to calculate Coulomb coupling by the sum over interaction between atomic transition charges approach of Howard et al.[49]. This approach have been shown to

reproduce very good values for the exciton interaction energy compared to point-dipole or extended-dipole approaches. Note that the magic angle of 55.4° is slightly different than that obtained in the case of the point-dipole approach (54.74°). This is because the NDI linker is not completely planar (see Supplementary Fig. 12) and we used the sum over interaction between atomic transition charges approach (see Supplementary Table 3) to evaluate the exciton interaction energy.

The ditopic cNDI linkers, irrespective of the size of attached "steric control units", form isoreticular Zn-SURMOF-2 structures, which have been reported in previous works[29,40]. In the previous work, also DFT calculations where the whole SURMOFs were described. With this previous information in hand, we have assembled the (geometry-optimized) ditopic linkers into a SURMOF-2 structure using an MOF constructor[27]. Test experiments have shown that the unit cell dimensions along the [100] direction and the [001] direction do not change upon substitution the cNDI linkers used here with R-groups. To predict the excitonic coupling (TDM–TDM coupling), the most important parameters are interchromophore distance and transition dipole moment geometry. Our proposed model structures, as described below, consider those two important parameters.

The proposed molecular dynamic (MD) scheme to predict the rotation angle $\theta$ for various R groups consists of 3 steps: (i) Optimization of the R-$NDI(OEt)_2$ monomer geometry based on optimized potential for liquid simulation (OPLS) force field[50–52] to get the dihedral angle ($\alpha$1) between the $NDI(OEt)_2$ plane and the phenyl plane (there is a second dihedral angle ($\alpha$2) between the phenyl group and the carboxylate group and this is basically zero for the isolated linker. In our discussion we focus on $\alpha$1); (ii) optimization of the interlayer distance ([010] direction) via calculating the potential energy of neighboring linkers (for a dimer) as a function of layer spacing (5.0–8.0 Å); (iii) construction and optimization of a trimer geometry (for final coordinates see SupplementaryTable 4) with the layer distance obtained in step (ii) and measure the rotation angle $\theta$ of central linker. MD simulations were carried out in GROMACS-2018.4 package[53–56].

For step (i) all initial R-$NDI(OEt)_2$ monomers were optimized using steepest descent method until the maximum force was smaller than 0.001 kJ mol$^{-1}$ nm$^{-1}$ or the maximum step 100,000 had been reached. Then, the backbone atoms (carboxylate groups, and atoms lying along the rotation axis) of the structures were fixed and remaining atoms were only allowed to rotate around the central axis. Next, simulated annealing simulations (SA-MD)[57] with constraint were carried out in step (i) and (iii) to find a global optimum geometry.

SA-MD were implemented using the following strategies: by coupling to the V-rescale thermostat under NVT ensemble (time step 1 fs), the system is quickly heating up from 300 to 1800 K in 25 ps to ensure random state and then stepwise

cooling down to 0 K at a cooling rate of $-2\,K\,ps^{-1}$. Finally, the monomer (step i) and trimers (step iii) structures obtained from the SA-MD were energy minimized using conjugated gradient method in 50,000 steps. Taking the optimized potential energy value as criteria, the geometry obtained via SA-MD is better than without SA-MD optimization.

The optical properties of NDI(OEt)$_2$ monomer and dimers were studied using the DFT method with the hybrid B3LYP functional[58]. For all atoms the triple-z-valence-plus-polarization (def2-TZVP) atomic orbital (AO) basis sets were used[59]. The Grimme D3 correction with Becke–Johnson damping was included for all systems studied[60,61]. The monomer NDI(OEt)$_2$ structure was fully optimized using gradient minimization techniques. The dimer structures for simulation of UV–Vis absorption spectra were constructed by placing the molecules 6.8 Å apart (MD optimized distance for 15 A [a trimer model], also confirmed by experimental data) to mimic the distance experimentally measured in the SURMOF and then both of them rotated by angles 0°, 34°, or 60° to simulate the steric effects of varying side groups (R). These angles were chosen based on the first level of theoretical simulations performed with the MDs on the trimers within GROMACS-2018.4 and reported in Supplementary Table 1. Calculations carried out at the TD-DFT level of theory also yielded a similar change in absorption spectra. They can be attributed to the Coulomb coupling between the neighboring cNDI cores of SURMOFs. To simulate UV–Vis absorption spectra 20 lowest lying singlet excitations have been calculated using time-dependent (TD) DFT approach (Supplementary Table 3).

The influence of different SCUs on NDI(OEt)$_2$ assemblies within SURMOF-2 was further verified using the DFT method, only for the experimentally fabricated SURMOFs. A trimer of NDI(OET)$_2$ was constructed and the distance between the linkers was set up according to experimental values. To improve the efficiency of simulations for larger systems, trimers were fully optimized using the resolution of identity (RI)-DFT procedure[62] together with multipole accelerated resolution of identity-J using the Perdew–Burke–Ernzerhof functional[63] and def2-SVP AO basis set[64]. We took three NDI(OEt)$_2$ organic linkers, placed them at the distance corresponding particular side group (R = H 5.8 Å, R = Me and Et 6.6 Å, R = iPr 6.8 Å) and fixed the terminal oxygen atoms to mimic the influence of zinc paddle-wheel (as in SURMOF-2). From these energy optimized trimers, only the central NDI(OEt)$_2$ has been visualized in Supplementary Fig. 5 (overlap of R = H and iPr). To increase the accuracy of the calculations Grimme D3 dispersion correction with Becke–Johnson damping and high-quality grid (m5) was also used. All DFT and TD-DFT calculations were performed in TURBOMOLE 7.3 software package[65].

**Optical characterization**. *Time-resolved spectroscopy*: For the time-resolved spectroscopy, time correlated single photon counting (TCSPC) with Nano LED light source (373 nm peak wavelength, 1 MHz max. repetition rate, 1.3 ns pulse duration) and FluoroHub Single Photon Detection Module was used.

PLQY measurements were performed according to the method described by de Mello et al.[66]. The beam of a 405 nm continuous wave laser diode (Thorlabs, DL5146-101S) was focused on the sample held inside an integrating sphere (Labsphere) with a diameter of 150 mm. For the detection two y-fiber-coupled uv/vis spectrometers (Avantes, AvaSpec-ULS-RS-TEC and Thorlabs, CCS200) were used. The spectral response of the whole detection system was calibrated using a calibration lamp (Ocean Optics, HL-3plus-INT-CAL).

**Fabrication of Zn-R-NDI(OEt)$_2$ SURMOF-2**. Ethanolic solutions of 1 mM zinc acetate and 20 μM R-NDI(OEt)$_2$ (R = methyl/ethyl/isopropyl) solutions (in ethanol) were sequentially deposited onto the precleaned quartz glass substrates using a spin-coating method in a lbl fashion. After the metal or linker deposition, the samples were washed with ethanol to remove unreacted metal/linker or by-products from the surface. For metal and linker both, the spin-coating time was fixed at 10 s with r.p.m. of 2000.

We have used SURMOF as a chromophore assembling template, instead of conventional MOF powders. The preference of SURMOFs over MOF powders was motivated by the requirements of optical and optoelectronic applications, which require optically active thin films deposited on transparent and conductive substrates with good control over the film thickness.

## Data availability
The data sets generated during and/or analyzed during the current study are available from the corresponding authors on reasonable request.

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

## Acknowledgements

R.H. acknowledges a postdoctoral fellowship from Alexander von Humboldt foundation. A.M., S.D. and F.O. acknowledge Région des Pays de la Loire through the program LUMOMAT for the financial support of this research with the project LumoMOF. M.J. acknowledges support from Karlsruhe School of Optics and Photonics (KSOP) graduate school. W.W. acknowledges support from the SFB 1176 "Structuring of Soft Matter". R. H., W.W. and C.W. acknowledge support through the Cluster "3DMM2O" funded by German DFG. I.A.H. acknowledges funding from DFG priority program SPP1928 COORNETs. All computations were performed on the computational resource ForHLR II funded by the Ministry of Science, Research and the Arts Baden-Württemberg and DFG.

## Author contributions

R.H., S.D., W.W., F.O. and C.W. conceived the idea and designed the experiments; A.M. and S.D. carried out the linker syntheses; R.H. and A.M. carried out the SURMOF syntheses; R.H. carried out the material characterizations, photophysical measurements and analyses with help from M.J., I.A.H. and B.S.R.; Q.Z., M.K., W.W. and D.J. planned and carried out the theoretical calculations; N.J. assisted in the design of a linker library; and R.H. prepared the manuscript with the inputs from all the co-authors.

## Additional information

**Competing interests:** The authors declare no competing interests.

