## [Peer Review File · Nature Communications]

Reviewers' comments:

Reviewer #1 (Remarks to the Author):

The manuscript reports NDI derived ligands for construction of MOFs with specific emphasis on systemic tuning of the spacial location of the NDI chromophores. They have successfully demonstrated that in this manuscript. As a result it was possible to predict the fluorescence behavior. This is a fundamental study and will make strong impact in the field not only for MOF but also for solution based self-assembly. cNDIs have huge potential and still they are under-explored and moreover the knowledge will help in future development of other chromophoric aggregates without quenching of fluorescence which is a common problem. The manuscript is written very well and conclusions are well supported by the data. It is recommended for publication.

Reviewer #2 (Remarks to the Author):

In their study, authors aimed to control (and avoid) the nonradiative decay processes caused by excitonic coupling of neighboring dye molecules in molecular solids, especially caused by cofacial π - π stacking. Using the construction paradigm of metalorganic frameworks (MOFs), they optimized chromophore packing in the solid state to avoid the often undesirable photoluminescence quenching. Here, Zn-SURMOF-2 was used as the basic molecular framework, using a layer-by-layer spin-coating method on hydroxyl-functionalized quartz and silicon substrates. In this MOF, a zinc-based paddle-wheel secondary building unit (SBU) is connected to a ditopic ligand serving as the chromophore. This produces a two-dimensional square grid assembly of tightly packed sheets. The chromophore packing in the MOF could be tuned (via rotation) by employing so-called "steric control units" (SCUs) in the framework, starting with the prototype bis-ethoxy-substituted-NDI (NDI(OEt)₂), with two phenyl carboxylate groups serving as ligands for metal coordination. The MOF resulting from this basic linker was characterized as a non-emissive H-aggregate.

Demonstrating their approach with core-substituted naphthalene diimides (NDIs), the authors used computational methods to discover the most favorable alignment of the chromophores to produce strongly emissive J-aggregates. In a next step, they built a library of "handle"-modified MOF chromophore linkers (modified at the phenyl carboxylate connectors) and, again using computational methods, searched for the SCUs best equipped to achieve the desired alignment. Finally, photophysical examination of a few selected, chemically synthesized MOFs (with SCUs -Me, -Et, -iPr, with interlayer distances between 0.58-0.68 nm) established the creation of J-aggregates featuring green emission with unusually high quantum yields for crystalline NDI-based solids.

This study establishes the interesting opportunities for tuning photophysical behavior of organic semiconductors offered by crystal design based upon the MOF construction paradigm.

Issues, questions:

- Callout SI Figure 5 should be SI Figure 6, please change the figure to show the twisted structure more clearly. Please also carefully check the other callouts for figures in the SI, there seem to be additional issues (e.g., 34° rotation calculation with MOs Fig. 7 is missing).

- In the introduction, please discuss the possible impact of chromophore quenching by certain constituent metal ions in the MOFs.

Summarizing, this is an interesting, well-written report on tuning optoelectronic properties through crystal design. It is recommended for publication after minor revision addressing the above points.

Reviewer #3 (Remarks to the Author):

This manuscript by Wöll et al. reports a joint theoretical and experimental study on luminescent MOFs. As a continued effort in the framework of an established KIT-CEISAM collaboration, the current work uses a MOF structure to assemble naphthalenediimide (NDI) chromophores in such a manner as to obtain luminescent materials. Using steric variations to control aggregation properties is certainly a classical but efficient approach. The attractive side of this work lies in the computational screening of a library of substituents, ahead of synthesis, to guide the design of MOFs displaying promising luminescence properties. As such, this work represents a significant progress in the field. However I have a major concern precluding acceptance of this manuscript, regarding the technical aspects of the theoretical contribution. The undertaken computational strategy is indeed appealing, but given the data reported in the main text and as Supporting Information, I am currently unable to say if the strategy is valid. In principle the 'Computational methods' section is meant to enable the reader to reproduce the calculations. With the given level of details, I doubt anyone could do so. Such a lack of rigour is deleterious, and a tremendous effort should be put in clarifying this. First and foremost, the only mention of calculations on a SURMOF structure (which would be required to claim reporting crystal structure prediction, CSP) is a sentence at the beginning of the article (lines 134-137 : 'Density functional theory (DFT) predicts a NDI(OEt)₂ geometry in the Zn-(NDI(OEt)₂) (A) SURMOF with a co-facial stacking of the NDI(OEt)₂ cores, leading to an undesirable H-aggregate type arrangement (See method section and Supporting Figure 2)'. However the method section says nothing about this calculation, and Supporting Figure 2 shows a trimer but no zinc. Elsewhere, only monomers, dimers or trimers of NDI are mentioned. To mimic SURMOF structures, computations on dimers or trimers with appropriate interchromophoric distances and tilt angles (these numbers being scarcely mentioned) could well be meaningful, but my gut feeling is that the confusion on what has exactly been computed sheds doubt on this validity. I am perfectly happy with using clever and sensible models, I just need to be convinced that the models are indeed sensible, and how they were computed. If I misunderstood what has actually been computed, i.e. if SURMOF structures 8A, 9A and 15A were indeed computed, then how do they compare with the experimental ones ? And could you please show them as Supporting Information ?

More detailed comments and questions can be found below, in order of appearance. There are quite a number of them, showing how unclear this manuscript can be. The most important ones call the line number in bold font.

To conclude, the concept and the computational strategy seem promising, but I am not yet convinced that the latter is fully valid. Nevertheless I am willing to reconsider a revised manuscript following major revision involving, above all, a real effort of clarification on what has been computed, why and how.

- Lines 63-66, 'This general interest has stimulated a large number of efforts to enforce J-type aggregation of chromophores into supramolecular assemblies.[11-13] However, very often the resulting aggregates do not have the desired properties, instead of high luminescence, nonradiative quenching processes result in "dark" materials.[4]'

Quoting references 11-13 is perfectly appropriate here but it is not correct to link them up with dark materials, since the materials reported in refs 11-13 are not dark.

- Lines 72-78, 'A typical example is the case of 9-anthracene carboxylic acid (ACA), a small, prototype chromophore. Non-substituted ACA crystallizes in a triclinic structure, with bright photoluminescent (PL) properties. When side groups are attached at the 10- position of ACA to modify the position of emission bands, the slightly modified chromophores crystallize in rather different crystal structures (triclinic, monoclinic, and orthorhombic).[17] As CSP methods struggle to predict the experimentally realized structures, researchers had to resort to trial-and-error strategies to optimize the photophysical properties of such ACA assemblies.'

Stressing on CSP seems misleading to me since this work does not report improved CSP approaches. Instead I would suggest to add something like '[serendipitous discovery] or extensive systematic screening[17]' at the end of line 71.

- Line 104 'rather close packing'

Ref 39 says 5.4 angstroms, could you please specify this number since it is used later as a reference ?

- Line 113 'for theta > 55.4°'

Could you please comment on this number with respect to the 54.7° magic angle (e.g. as in ref. 5) ?

- Line 113 'the Coulomb coupling'

The signs of the Coulomb coupling JC and the charge-transfer coupling JCT can vary independently. Could you please comment on the possible interference between Coulomb and CT couplings (e.g. J. Chem. Phys. 2015, 143, 244707) ? Why do you think JC is sufficient to predict the type of aggregate ? In other words, why would Kasha's seminal exciton model hold here ?

- Line 119 'assemble MOFs in silico'

This is not consistent with the computational methods (more details are given below)

- Figure 1

Please keep the same colour code throughout.

Fig 1a : Ph are blue, zinc is green ; Fig 1b : Ph are green

Fig 1a : R groups are red ; Fig 2a : R groups are blue

- Figure 1b

* 'TDM-TDM coupling': please expand TDM acronym once (or in text line 140)

* caption 'Coulomb coupling vs rotation angle theta (as shown in Figure 1a at the right hand side)': theta is in fact shown in the inset of Figure 1b

* what does the green line correspond to ?

- Line 128 'Crystal structure prediction of NDI(OEt2)-assembly'

Again if the calculations are not made on SURMOF structures, this title is misleading. Something like 'Structural prediction of a library of NDI assemblies' would be more correct, without diminishing the quality or the impact of the work.

- Line 135 'co-facial stacking of the cores'

Please report the interchromophoric distance (in the text or on Supporting Figure 2)

- Line 141 'closely spaced chromophores'

Please report the interchromophoric distance

- Line 143 'placed at a distance (x) of 6.8 angstroms'

Is that what 'closely spaced' means ? This is significantly larger than the 5.4 angstroms distance of ref 39. Where does this number come from ?

- Line 145 'for theta > 55.4°'

Where does this number come from ? Isn't there a plot missing ? Second half of Supporting Figure 3 'angle dependence of monomers', Jmn vs theta ?

- Line 158 'apparently more bulky groups yielded a smaller angle of theta'

In the methyl/benzyl comparison, this sentence should mention that they both induce similar dihedral angles (according to Supporting Table 1). Otherwise the comparison does not hold.

- Line 158 'methyl (9)'
In Figure 2a, methyl is 8

- Line 159 'benzyl (2)'
In Figure 2a, benzyl is 6

- Line 167, figure 2b
I understand that the rotation angles were obtained from Gromacs-optimized trimers, is this correct ? If so, this should be clarified.

- Line 195 'Supporting Information Figure 5'
is in fact Supporting Figure 6

- Line 204 'Supporting Information Figure 6'
is in fact Supporting Figure 5

- Line 207 'vibronic overtones at 436 and 472 nm'
Does the vibrational spacing fit with a specific vibrational mode ? Which one ? Does it also fit with the simulated spectra shown in Supporting Figure 7 ?

- Line 212 'We noted that the coupling energy for A and 9A or 15A did not differ markedly (Figure 1b, ~50 meV)'

The data for R=iPr (as in SURMOF 15A) is shown on Figure 1b and is slightly negative (roughly -20meV) ; the data for R=Et (as in SURMOF 9A) is shown on Figure 1b and is slightly positive (roughly 20 meV) ; the data for R=H (as in SURMOF A) is not shown on Figure 1b. This doesn't make sense.

- Line 216 'To explain this behavior we employed time dependent density functional theory (TD-DFT) to simulate the electronic absorption spectra of the NDI(OEt)₂ dimers with rotation angles 34 and 60° for H and J-type coupling, respectively (Supporting Information Figure 7 and 8)'

* How were these angles chosen ? The angles reported in Supporting Table 1 for R=Me or Et and R=iPr could have been chosen advantageously. Also see the vague expression 'certain angle' line 317.

* What was the interchromophoric distance in these TD-DFT calculations ? Also see the vague expression 'proper distance' line 316.

* What is the origin of the additional red shoulder in the computed absorption spectrum of the 60°-tilted dimer ? If this shoulder is of vibronic origin, a vibrationally-resolved spectrum should be computed to confirm it. If, on the other hand, this shoulder is due to an additional electronic transition, it would be interesting to identify it.

- Line 223

Why are 8A and 9A so different in absorption (Figure 4a) but so similar in emission (Figure 4b,c) and structure (Supporting Table 1) ?

- Figure 4a

The colour code seems wrong with an inversion between 8A and 9A (inconsistency with the text and with the caption of Figure 3)

- Figure 4b

The partly hidden caption should be completely hidden

- Figure 4d

* Please provide the emission spectra corresponding to Figure 4d-inset(ii) as Supporting Information

* are the TDM-TDM coupling values shown on Figure 4d supposed to match some of those shown on Figure 1b ? It is not clear to me what the difference between these two sets of data is.

- Line 264 'crystal structure prediction'

Same remark as above : at this stage I can't say if and how any SURMOF structure was computed at all.

- Line 267 '+40 meV'

* Was this computed for a dimer with R=H or for SURMOF A ?

* Similarly, '-15 meV', was this computed for a dimer with R=iPr or for SURMOF 15A ?

- Line 293 'Optimization of the layer distance via calculating the potential energy of neighboring linkers as a function of layer spacing (5.0~8.0 angstroms)'

Is this performed on dimers or on MOF structures ? Since step (i) concerns monomers and step (iii) concerns trimers, I would rather think that step (ii) concerns dimers, in which case there are no calculations on SURMOF structures.

- Line 307 'Taking the optimized potential energy value as criteria, the geometry obtained via SA-MD is substantially better than without SA-MD optimization'

Please give elements of comparison (distances, energies).

- Line 316 'proper distance'

For clarity, please recall these numbers for each R group.

- Line 317 'certain angle'

For clarity, please recall these numbers for each R group.

- Line 326 'Becke D3 dispersion'

should be 'Grimme D3 dispersion'.

- Line 327 'high quality grid'

Please be more specific. Also please specify which program was used for the DFT and TD-DFT calculations.

- Line 391 'photochemical'

should be 'photomechanical'.

- Supporting Information Table 1

If I understood correctly, these calculations were performed on Gromacs-optimized trimers. For clarity this information could be recalled in the Table caption.

- Supporting Information Figure 2

* Please specify interchromophoric distance

* If the calculation indeed concerns a SURMOF structure, how was it performed and where are the zinc ions ?

- Supporting Information Figure 3

The angle dependence of the Coulomb coupling should be shown.

- Supporting Information line 151 'The shift in (010) diffraction from A to the other SURMOFs indicate a slight changes (~ 0.1 angstrom) in the inter-sheet distances'

Computed intermolecular distances given in Supporting Table 1 (presumably in trimers, not in SURMOFs) show that the intermolecular distances vary by 1 angstrom (from 5.8 angstroms for R=H to 6.8 angstroms for R=iPr). Isn't this a concern to you that this doesn't fit with the experimental intermolecular distances, which are the same for all SURMOFs ?

- Supporting Information line 155

For clarity please specify 'distances as in respective experimental SURMOF structures'.

- Supporting Information Figure 6

* The caption says 'DFT simulated geometries of A and 15A', which is very confusing with respect to the Computational methods section. Is there any zinc in these calculations ? It seems as if these calculations were performed on monomers.

* According to Supporting Table 1, the dihedral angle between the NDI core and the Ph substituents is 69.1° for R=H and 87.4° for R=iPr. This is not obvious from Supporting Figure 6.

- Supporting Information line 172

* Please specify interchromophoric distance in the dimers.

* Were they optimized as in Gromacs step (ii) ?

- Supporting Information Figure 8

* Please specify interchromophoric distance in the dimer.

* Please show 60° angle on the figure. Is this the theta angle of Figure 1b ?

* Is the image at the left of the computed absorption spectrum a sort of top view of the one shown in the inset of the spectrum ?

* Please show the full spectrum (the red region is currently cut)

* 'top' and 'bottom' do not correspond to the current display

Point-by-point response

Reviewer #1 (Remarks to the Author):

The manuscript reports NDI derived ligands for construction of MOFs with specific emphasis on systemic tuning of the spacial location of the NDI chromophores. They have successfully demonstrated that in this manuscript. As a result it was possible to predict the fluorescence behavior. This is a fundamental study and will make strong impact in the field not only for MOF but also for solution based self-assembly. cNDIs have huge potential and still they are under-explored and moreover the knowledge will help in future development of other chromophoric aggregates without quenching of fluorescence which is a common problem. The manuscript is written very well and conclusions are well supported by the data. It is recommended for publication.

Response: We are delighted to learn that the reviewer recognized the importance of our work, and recommends its publication as is.

Reviewer #2 (Remarks to the Author):

In their study, authors aimed to control (and avoid) the nonradiative decay processes caused by excitonic coupling of neighboring dye molecules in molecular solids, especially caused by cofacial π - π stacking. Using the construction paradigm of metalorganic frameworks (MOFs), they optimized chromophore packing in the solid state to avoid the often undesirable photoluminescence quenching. Here, Zn-SURMOF-2 was used as the basic molecular framework, using a layer-by-layer spin-coating method on hydroxyl-functionalized quartz and silicon substrates. In this MOF, a zinc-based paddle-wheel secondary building unit (SBU) is connected to a ditopic ligand serving as the chromophore. This produces a two-dimensional square grid assembly of tightly packed sheets. The chromophore packing in the MOF could be tuned (via rotation) by employing so-called “steric control units” (SCUs) in the framework, starting with the prototype bis-ethoxy-substituted-NDI (NDI(OEt)₂), with two phenyl carboxylate groups serving as ligands for metal coordination. The MOF resulting from this basic linker was characterized as a non-emissive H-aggregate.

Demonstrating their approach with core-substituted naphthalene diimides (NDIs), the authors used computational methods to discover the most favorable alignment of the chromophores to produce strongly emissive J-aggregates. In a next step, they built a library of “handle”-modified MOF chromophore linkers (modified at the phenyl carboxylate connectors) and, again using computational methods, searched for the SCUs best equipped to achieve the desired alignment. Finally, photophysical examination of a few selected, chemically synthesized MOFs (with SCUs

-Me, -Et, -iPr, with interlayer distances between 0.58-0.68 nm) established the creation of J-aggregates featuring green emission with unusually high quantum yields for crystalline NDI-based solids.

This study establishes the interesting opportunities for tuning photophysical behavior of organic semiconductors offered by crystal design based upon the MOF construction paradigm.

Issues, questions:

- Callout SI Figure 5 should be SI Figure 6, please change the figure to show the twisted structure more clearly. Please also carefully check the other callouts for figures in the SI, there seem to be additional issues (e.g., 34° rotation calculation with MOs Fig. 7 is missing).

Response: We have corrected the figure numbers and included a comprehensive model to show the twisted structure. Other missing information have also been included and marked in yellow for convenient reading. We thank the reviewer for pointing out these issues.

- In the introduction, please discuss the possible impact of chromophore quenching by certain constituent metal ions in the MOFs.

Response: In following the suggestion of the reviewer, we have included a sentence with reference to metal ion based quenching in the introduction of the revised manuscript.

Summarizing, this is an interesting, well-written report on tuning optoelectronic properties through crystal design. It is recommended for publication after minor revision addressing the above points.

Response: We are grateful to the reviewer for his/her appreciation of our approach, the very valuable comments on the manuscript, and for the recommendation to publish this paper in Nature Communications.

Reviewer #3 (Remarks to the Author):

This manuscript by Wöll et al. reports a joint theoretical and experimental study on luminescent MOFs. As a continued effort in the framework of an established KIT-CEISAM collaboration, the current work uses a MOF structure to assemble naphthalenediimide (NDI) chromophores in such a manner as to obtain luminescent materials. Using steric variations to control aggregation properties is certainly a classical but efficient approach. The attractive side of this work lies in the computational screening of a library of substituents, ahead of synthesis, to guide the design of MOFs displaying promising luminescence properties. As such, this work represents a significant progress in the field. However I have a major concern precluding acceptance of this manuscript, regarding the technical aspects of the theoretical contribution. The undertaken computational strategy is indeed appealing, but given the data reported in the main text and as Supporting Information, I am currently unable to say if the strategy is valid. In principle the 'Computational methods' section is meant to enable the reader to reproduce the calculations. With the given level of details, I doubt anyone could do so. Such a lack of rigour is deleterious, and a tremendous effort should be put in clarifying this. First and foremost, the only mention of calculations on a SURMOF structure (which would be required to claim reporting crystal structure prediction, CSP) is a sentence at the beginning of the article (lines 134-137 : 'Density functional theory (DFT) predicts a NDI(OEt)₂ geometry in the Zn-(NDI(OEt)₂) (A) SURMOF with a co-facial stacking of the NDI(OEt)₂ cores, leading to an undesirable H-aggregate type arrangement (See method section and Supporting Figure 2)'. However the method section says nothing about this calculation, and Supporting Figure 2 shows a trimer but no zinc. Elsewhere, only monomers, dimers or trimers of NDI are mentioned. To mimic SURMOF structures, computations on dimers or trimers with appropriate interchromophoric distances and tilt angles (these numbers being scarcely mentioned) could well be meaningful, but my gut feeling is that the confusion on what has exactly been computed sheds doubt on this validity. I am perfectly happy with using clever and sensible models, I just need to be convinced that the models are indeed sensible, and how they were computed. If I misunderstood what has actually been computed, i.e. if SURMOF structures 8A, 9A and 15A were indeed computed, then how do they compare with the experimental ones ? And could you please show them as Supporting Information?

Response: We first thank this reviewer for identifying our work as a significant progress in the field. We fully accept the request of this reviewer for providing additional details on the computational work reported on in our paper. We have subjected the computational method section to a major revision and now provide a much more detailed presentation of our “crystal structure prediction”, or CSP, methodology. Some additional information has also been added to the supporting information.

The “CSP” approach is briefly described here: The ditopic cNDI-linkers, irrespective of the size of attached “steric control units”, form isorecticular Zn-SURMOF-2 structures, which have been reported in previous works. In these previous works, also DFT-calculations where the

whole SURMOFs were optimized have been described (DOI: 10.1038/srep00921; DOI: 10.1002/chem.201702968). With this previous information in hand, we have assembled the (geometry-optimized) ditopic linkers into a SURMOF-2 structure using a MOF constructor (DOI: 10.1021/jp507643v; Ref. 27 in the manuscript). Test experiments have shown that the unit cell dimensions along the [100]-direction and the [001] direction do not change upon substitution the cNDI-linkers used here with R-groups, so the same dimensions were used in all calculations. The unit cell length along the [010]-direction has been determined by calculations for a dimer.

A more detailed description of these calculations has been added to the method section of the revised manuscript. We believe that the present version of the “CSP” methodology is comprehensive and answers all the queries of the reviewer.

We would like to stress that the message of the paper is that this approximate method is indeed helpful to find a functional group R for which the initially dark SURMOF is turned into a highly emissive one. The goal of the paper is not to provide a through theoretical description of all the different SURMOF structures considered.

More detailed comments and questions can be found below, in order of appearance. There are quite a number of them, showing how unclear this manuscript can be. The most important ones call the line number in bold font.

To conclude, the concept and the computational strategy seem promising, but I am not yet convinced that the latter is fully valid. Nevertheless I am willing to reconsider a revised manuscript following major revision involving, above all, a real effort of clarification on what has been computed, why and how.

Response: Again, we accept the criticism of the referee and have considered every single critical item in his/her list. Clearly, these informations are crucial to the concept presented here and adding a more detailed discussion has substantially improved the quality of our manuscript. In the revised manuscript, we have included every important detail of the computational methods. Below, we provide answers to all questions raised in a point-by-point fashion.

- Lines 63-66, 'This general interest has stimulated a large number of efforts to enforce J-type aggregation of chromophores into supramolecular assemblies.[11-13] However, very often the resulting aggregates do not have the desired properties, instead of high luminescence, nonradiative quenching processes result in “dark” materials.[4]'

Quoting references 11-13 is perfectly appropriate here but it is not correct to link them up with dark materials, since the materials reported in refs 11-13 are not dark.

Response: We do not understand this criticism. In fact, we did not link Ref. 11-13 to dark materials. Only Ref. [4] is linked to such non-luminescent assemblies. Therefore, we feel that no changes are required in this case.

- Lines 72-78, 'A typical example is the case of 9-anthracene carboxylic acid (ACA), a small, prototype chromophore. Non-substituted ACA crystallizes in a triclinic structure, with bright photoluminescent (PL) properties. When side groups are attached at the 10- position of ACA to modify the position of emission bands, the slightly modified chromophores crystallize in rather different crystal structures (triclinic, monoclinic, and orthorhombic).[17] As CSP methods struggle to predict the experimentally realized structures, researchers had to resort to trial-and-error strategies to optimize the photophysical properties of such ACA assemblies.'

Stressing on CSP seems misleading to me since this work does not report improved CSP approaches. Instead I would suggest to add something like '[serendipitous discovery] or extensive systematic screening[17]' at the end of line 71.

Response: We do not agree with the referee. Indeed, what we report is the prediction of crystal structures by using an approximate scheme as described above. Reducing the number of degrees of freedom to essentially one (rotation of the cNDI around linker axis) makes CSP much easier, and this is why our approach was successful. This is a central point of the paper, and we would like to stay with the original wording.

- Line 104 'rather close packing'

Ref 39 says 5.4 angstroms, could you please specify this number since it is used later as a reference?

Response: Actually, the value 5.4 Å relates to a SURMOF with a different cNDI-linker. The cited paper (ref 39) demonstrated that for inter-sheet distance of ~ 5.4 Å strong excitonic couplings are present among the chromophores. Please note, that for the NDI(OEt)₂ linker used in the present SURMOFs with R=H the inter-chromophore distance is 5.8 Å, and this has been clearly mentioned in the manuscript.

- **Line 113 'for theta > 55.4°**

Could you please comment on this number with respect to the 54.7° magic angle (e.g. as in ref. 5)?

Response: The magic angle of 54.7° is obtained when using a simple dipole-dipole approximation. In our calculations, we used the more sophisticated TrEsp approach, where transition charges were fitted to every atom in our monomer from electrostatic potential fit (as in reference 43). In fact, the monomer used to calculate the transition electrostatic potential is modified in the DFT optimization of the organic linkers, and the final structure is not completely planar, (see last part of SI where all coordinates and partition charges used in Coulomb coupling

calculations are given). As a result, the final angle is slightly different from the magic angle obtained from simpler models.

- Line 113 'the Coulomb coupling'

The signs of the Coulomb coupling JC and the charge-transfer coupling JCT can vary independently.

Could you please comment on the possible interference between Coulomb and CT couplings (e.g. J. Chem. Phys. 2015, 143, 244707) ? Why do you think JC is sufficient to predict the type of aggregate ? In other words, why would Kasha's seminal exciton model hold here ?

Response: Charge-transfer coupling (JCT) is usually present within “tightly stacked” organic systems with small (below 4 Å) intermolecular distances. Since in our case the minimal distance between the cNDI cores is much larger, namely 5.8 Å for R=H, and increases up to 6.8 Å in the case of the R=iPr, we conclude that the charge-transfer couplings will be small, in fact much smaller than the Coulomb coupling between the neighboring chromophores. This expectation is confirmed by the inspection of HOMO and LUMO orbital overlaps, which are small. Altogether these observations give us confidence that we can base our theoretical explanation only on dominating Coulomb coupling.

Line 119 'assemble MOFs in silico'

This is not consistent with the computational methods (more details are MOF given below)

Response: As already stated above, we used a MOF constructor developed by some of us earlier (DOI: 10.1021/jp507643v; Ref. 27 in the manuscript) to take ditopic linkers and construct a SURMOF-2 structure. We believe that for this process the wording “assemble MOFs in silico” is appropriate. We have added an additional reference to make this point more clear.

- Figure 1

Please keep the same colour code throughout.

Fig 1a : Ph are blue, zinc is green ; Fig 1b : Ph are green

Response: We have modified Fig. 1 – now the color scheme should be consistent.

Fig 1a : R groups are red ; Fig 2a : R groups are blue

Response: These problems have been removed.

- Figure 1b

* 'TDM-TDM coupling': please expand TDM acronym once (or in text line 140)

Response: We have added the expanded acronym in text line 140.

* caption 'Coulomb coupling vs rotation angle theta (as shown in Figure 1a at the right hand side)': theta is in fact shown in the inset of Figure 1b

Response: This problem has been removed.

* what does the green line correspond to ?

Response: The green line corresponds to the calculated TDM-TDM couplings for different rotation of NDI(OEt)₂, as mentioned in the figure caption.

Line 128 'Crystal structure prediction of NDI(OEt₂)-assembly'

Again if the calculations are not made on SURMOF structures, this title is misleading. Something like 'Structural prediction of a library of NDI assemblies' would be more correct, without diminishing the quality or the impact of the work.

Response: As mentioned before, we have used a MOF-constructor to assemble a SURMOF-2 from the ditopic linkers and then used MD-simulations to optimize the unit cell size along the [010] crystallographic direction. We agree that this is not a full structure determination. Following the suggestion of the reviewer, we have revised the line 128 as follows: "Structural prediction of a library of NDI(OEt)₂ assemblies".

- Line 135 'co-facial stacking of the cores'

Please report the interchromophoric distance (in the text or on Supporting Figure 2)

Response: The missing information has been added to text.

- Line 141 'closely spaced chromophores'

Please report the interchromophoric distance

Response: The missing information has been added to text.

- Line 143 'placed at a distance (x) of 6.8 angstroms'
Is that what 'closely spaced' means ? This is significantly larger than the 5.4 angstroms distance of ref 39.

Where does this number come from ?

Response: This number has been obtained by determining the [010] unit cell parameter from the MD calculations for a non-rotated (R=H) cNDI linker. This value is larger than the 5.4 Å reported previously since the linkers are quite different..

Line 145 'for theta > 55.4°'

Where does this number come from? Isn't there a plot missing ? Second half of Supporting Figure 3 'angle dependence of monomers', J_{mn} vs theta ?

Response: This number comes from Figure 1b where the angle dependence of the Coulomb coupling for a fixed interchromophoric distance of 6.8 Å is displayed (green line). The same plot is now also introduced in the Supporting Figure 3.

- Line 158 'apparently more bulky groups yielded a smaller angle of theta'
In the methyl/benzyl comparison, this sentence should mention that they both induce similar dihedral angles (according to Supporting Table 1). Otherwise the comparison does not hold.

Response: This problem has been fixed.

- Line 158 'methyl (9)'
In Figure 2a, methyl is 8

Response: This problem has been fixed.

Line 159 'benzyl (2)'

In Figure 2a, benzyl is 6

Response: This problem has been fixed.

- Line 167, figure 2b

I understand that the rotation angles were obtained from Gromacs-optimized trimers, is this correct? If so, this should be clarified.

Response: Yes, those were obtained from GROMACS. We have augmented the text of the figure caption to make this point clear.

- Line 195 'Supporting Information Figure 5' is in fact Supporting Figure 6

Response: The supporting figures have been revised accordingly.

- Line 204 'Supporting Information Figure 6' is in fact Supporting Figure 5

Response: The supporting figures have been revised accordingly.

Line 207 'vibronic overtones at 436 and 472 nm'

Does the vibrational spacing fit with a specific vibrational mode? Which one? Does it also fit with the simulated spectra shown in Supporting Figure 7?

Response: Yes, the vibronic features in the solvated cNDI-units can be related to vibrational modes, as has been discussed previously in the literature (DOI: 10.1039/c0cc00078g). However, this is not relevant for our work, since the vibronic features changed in the corresponding SURMOF data (as a result of strong excitonic coupling, see DOI:10.1002/anie.201708267).

Since vibronic couplings were not included in the simulations, a direct comparison is not meaningful here. Please note also, that the absorption bands are shifted in comparison to experimental values, evidently due to the presence of solvation effects in the experimental results for the solvated monomers.

Line 212 'We noted that the coupling energy for A and 9A or 15A did not differ markedly (Figure 1b, ~50 meV)'

The data for R=iPr (as in SURMOF 15A) is shown on Figure 1b and is slightly negative (roughly -20meV) ; the data for R=Et (as in SURMOF 9A) is shown on Figure 1b and is slightly positive (roughly 20 meV) ; the data for R=H (as in SURMOF A) is not shown on Figure 1b. This doesn't make sense.

Response: The referee is correct. We have included the predicted θ for R=H by GROMAC-2018.4 in Figure 1b.

Line 216 'To explain this behavior we employed time dependent density functional theory (TD-DFT) to simulate the electronic absorption spectra of the NDI(OEt)₂ dimers with rotation angles 34 and 60° for H and J-type coupling, respectively (Supporting Information Figure 7 and 8)'

* How were these angles chosen ? The angles reported in Supporting Table 1 for R=Me or Et and R=iPr could have been chosen advantageously. Also see the vague expression 'certain angle' line 317.

Response: We are grateful for this comment by the referee. Briefly, we took two rotation angles (34 and 60 °), which clearly represent different aggregation types, H and J, respectively. A corresponding statement has been added to the manuscript.

* What was the interchromophoric distance in these TD-DFT calculations? Also see the vague expression 'proper distance' line 316.

Response: We have added the requested information (6.8 Å) to the manuscript.

* What is the origin of the additional red shoulder in the computed absorption spectrum of the 60°-tilted dimer ? If this shoulder is of vibronic origin, a vibrationally-resolved spectrum should be computed to confirm it. If, on the other hand, this shoulder is due to an additional electronic transition, it would be interesting to identify it.

Response: The origin of the additional transition at 485.62 nm in the absorption spectrum is an electronic transition. The known fact is that due to the parallel transition dipole orientations

within coordinated chromophores in close spatial proximity low lying transitions are forbidden in the case of non-emissive H-aggregates. However, when two chromophores form a J-aggregate and two transition dipoles become almost collinear, this dark state becomes allowed and low lying visible absorption peaks are observed in the spectrum (DOI: 10.1039/C0DT01226B). From the comparison of the absorption spectra for the case of 34° and 60° in the Supporting Figure 8 (c and d), there is a visible rise in intensity of the previously dark singlet transition S1. From the analysis of the leading contributions for S1 transition it can be seen that it is dominated by HOMO-LUMO transition localized on central part of both chromophores.

Line 223

Why are 8A and 9A so different in absorption (Figure 4a) but so similar in emission (Figure 4b,c) and structure (Supporting Table 1) ?

Response: 8A and 9A absorption spectra are different because their transition dipole moment (TDM) couplings are different (in line with the theoretical predictions). Although the steady state emission spectra (Figure 4b) of 8A and 9A do not differ significantly, there is an evident difference in PLQY (Figure 4d) for these two species. A more detailed answers to this questions would require a thorough theoretical analysis, which we feel is beyond the scope of the present paper. In fact, the message of the paper is that we use computational screening to find side groups which turn the “dark” SURMOF into an emissive one. The success of this approach is evident from the experimental observation that the SURMOF fabricated from the optimized linkers indeed is highly emissive.

- Figure 4a

The colour code seems wrong with an inversion between 8A and 9A (inconsistency with the text and with the caption of Figure 3)

Response: We are grateful to the referee for pointing out this problem. Figure 3 has been corrected and the color codes are now consistent.

- Figure 4b

The partly hidden caption should be completely hidden

Response: Nothing is hidden in Figure 4b, it is a relevant legend.

Figure 4d

* Please provide the emission spectra corresponding to Figure 4d-inset(ii) as Supporting Information

Response: We have followed the suggestion of the referee and have added the requested material to the supporting information.

* are the TDM-TDM coupling values shown on Figure 4d supposed to match some of those shown on Figure 1b ? It is not clear to me what the difference between these two sets of data is.

Response: We thank the reviewer for pointing out this issue. We have chosen the TDM-TDM coupling values from figure 1b, which were computed for the initially predicted rotation angles of **8A**, **9A** and **15A**. Figure 4d shows a correlation of prediction and experimental results. In the revised figure 4d, the values can be correlated perfectly with figure 1b and supporting table 1. We have also mentioned in the figure caption of figure 4d, that the TDM-TDM couplings plotted are from Figure 1b.

Line 264 'crystal structure prediction'

Same remark as above : at this stage I can't say if and how any SURMOF structure was computed at all.

Response: As explained above we feel that using this term is justified. We have used approximate methods to predict SURMOF structures, we did not claim that the SURMOF structure was determined using a full DFT structure optimization.

Line 267 '+40 meV'

* Was this computed for a dimer with R=H or for SURMOF A?

Response: This was computed for a dimer with R=H.

* Similarly, '-15 meV', was this computed for a dimer with R=iPr or for SURMOF 15A?

Response: This was computed for a dimer with R=iPr.

Line 293 'Optimization of the layer distance via calculating the potential energy of neighboring linkers as a function of layer spacing (5.0~8.0 angstroms)'

Is this performed on dimers or on MOF structures? Since step (i) concerns monomers and step (iii) concerns trimers, I would rather think that step (ii) concerns dimers, in which case there are no calculations on SURMOF structures.

Response: This optimization was performed on a dimer of cNDI with related SCU attached to it.

- Line 307 'Taking the optimized potential energy value as criteria, the geometry obtained via SA-MD is substantially better than without SA-MD optimization'

Please give elements of comparison (distances, energies).

Response: Following is a comparison of potential energy values obtained with and without SA-MD: By Using SA-MD optimization, for linker **13**, the monomer (step i) potential energy value decreased by 32 kJ/mol, and for trimer (step iii) it is 53 kJ/mol.

- Line 316 'proper distance'

For clarity, please recall these numbers for each R group.

Response: The requested information has been added to the manuscript.

- Line 317 'certain angle'

For clarity, please recall these numbers for each R group.

Response: The requested information has been added to the manuscript.

- Line 326 'Becke D3 dispersion'

should be 'Grimme D3 dispersion'.

Response: We have changed the wording in accord with this suggestion.

- Line 327 'high quality grid'

Please be more specific. Also please specify which program was used for the DFT and TD-DFT calculations.

Response: We have added the requested information to the paper.

- Line 391 'photochemical' should be 'photomechanical'.

Response: Has been corrected.

- Supporting Information Table 1

If I understood correctly, these calculations were performed on Gromacs-optimized trimers. For clarity this information could be recalled in the Table caption.

Response: We have added the requested information to the manuscript.

Supporting Information Figure 2

* Please specify interchromophoric distance

* If the calculation indeed concerns a SURMOF structure, how was it performed and where are the zinc ions ?

Response: The values for the inter-chromophore distance have been added to the manuscript. The calculations were performed for a trimer model, without considering the zinc paddle-wheel units. This information has been added to the legend of the figure.

Supporting Information Figure 3

The angle dependence of the Coulomb coupling should be shown.

Response: Please note that the angle dependence is already present Figure 1b of the original manuscript. In order to account for the criticism of the referee, we have added this information to Figure 3 (bottom).

Supporting Information line 151 'The shift in (010) diffraction from A to the other SURMOFs indicate a slight changes (~ 0.1 angstrom) in the inter-sheet distances'
Computed intermolecular distances given in Supporting Table 1 (presumably in trimers, not in SURMOFs) show that the intermolecular distances vary by 1 angstrom (from 5.8 angstroms for R=H to 6.8 angstroms for R=iPr). Isn't this a concern to you that this doesn't fit with the experimental intermolecular distances, which are the same for all SURMOFs?

Response: We are grateful to the referee for pointing out this problem. The sentence has been revised. From A to 15A, the change in inter-sheet distance is $\sim 1 \text{ \AA}$, but not $\sim 0.1 \text{ \AA}$ (also mentioned in the main text). So the calculation and experimental values agree well.

- Supporting Information line 155

For clarity please specify 'distances as in respective experimental SURMOF structures'.

Response: We have modified the manuscript in accord with this suggestion.

Supporting Information Figure 6

* The caption says 'DFT simulated geometries of A and 15A', which is very confusing with respect to the Computational methods section. Is there any zinc in these calculations? It seems as if these calculations were performed on monomers.

Response: We have modified the manuscript in accord with this suggestion..

* According to Supporting Table 1, the dihedral angle between the NDI core and the Ph substituents is 69.1° for R=H and 87.4° for R=iPr. This is not obvious from Supporting Figure 6.

Response: We have modified the manuscript in accord with this suggestion.

- Supporting Information line 172

* Please specify interchromophoric distance in the dimers.

Response: We have modified the manuscript in accord with this suggestion. In the revised version these distances are mentioned.

* Were they optimized as in Gromacs step (ii) ?

Response: Yes, we obtained this from GROMACS-2018.4 based calculations (in step ii).

- Supporting Information Figure 8

* Please specify interchromophoric distance in the dimer.

Response: Now the distances are mentioned.

* Please show 60° angle on the figure. Is this the theta angle of Figure 1b?

Response: Now we have added this in the revised version.

Unfortunately, we do not understand the question “Is this the theta angle of Figure 1b?”. Figure 1b and SI Figure 8 are clearly described in the corresponding figure captions.

* Is the image at the left of the computed absorption spectrum a sort of top view of the one shown in the inset of the spectrum ?

Response: Yes, it is the top view. It is described in the revised version.

* Please show the full spectrum (the red region is currently cut)

Response: The red parts of the spectra were deliberately omitted due to the fact that there are no electronic transitions. The additional Table 3 in the Supporting Information is now added, with the overview of the all calculated transitions and corresponding wavelengths. This clearly shows that lowest singlet transitions for all chromophore assemblies lay below 500 nm.

* 'top' and 'bottom' do not correspond to the current display

Response: We have modified the manuscript in accord with this suggestion.

Reviewers' comments:

Reviewer #3 (Remarks to the Author):

A number of red lights have turned yellow or green with this revised version, but a few red lights remain. Please reconsider the points written in red (major) and highlighted in yellow (minor) in the attached pdf file. I know there are strong space constraints, but there are ways to ensure that the space constraints aren't detrimental to science.

See below review for the detailed comments and queries

Point-by-point response

A number of red lights have turned yellow or green with this revised version, but a few red lights remain. Please reconsider the points written in red (major) and highlighted in yellow (minor). I know there are strong space constraints, but there are ways to ensure that the space constraints aren't detrimental to science.

Reviewer #3 (Remarks to the Author):

This manuscript by Wöll et al. reports a joint theoretical and experimental study on luminescent MOFs. As a continued effort in the framework of an established KIT-CEISAM collaboration, the current work uses a MOF structure to assemble naphthalenediimide (NDI) chromophores in such a manner as to obtain luminescent materials. Using steric variations to control aggregation properties is certainly a classical but efficient approach. The attractive side of this work lies in the computational screening of a library of substituents, ahead of synthesis, to guide the design of MOFs displaying promising luminescence properties. As such, this work represents a significant progress in the field. However I have a major concern precluding acceptance of this manuscript, regarding the technical aspects of the theoretical contribution. The undertaken computational strategy is indeed appealing, but given the data reported in the main text and as Supporting Information, I am currently unable to say if the strategy is valid. In principle the 'Computational methods' section is meant to enable the reader to reproduce the calculations. With the given level of details, I doubt anyone could do so. Such a lack of rigour is deleterious, and a tremendous effort should be put in clarifying this. First and foremost, the only mention of calculations on a SURMOF structure (which would be required to claim reporting crystal structure prediction, CSP) is a sentence at the beginning of the article (lines 134-137 : 'Density functional theory (DFT) predicts a NDI(OEt)₂ geometry in the Zn-(NDI(OEt)₂) (A) SURMOF with a co-facial stacking of the NDI(OEt)₂ cores, leading to an undesirable H-aggregate type arrangement (See method section and Supporting Figure 2)'. However the method section says nothing about this calculation, and Supporting Figure 2 shows a trimer but no zinc. Elsewhere, only monomers, dimers or trimers of NDI are mentioned. To mimic SURMOF structures, computations on dimers or trimers with appropriate interchromophoric distances and tilt angles (these numbers being scarcely mentioned) could well be meaningful, but my gut feeling is that the confusion on what has exactly been computed sheds doubt on this validity. I am perfectly happy with using clever and sensible models, I just need to be convinced that the models are indeed sensible, and how they were computed. If I misunderstood what has actually been computed, i.e. if SURMOF structures 8A, 9A and 15A were indeed computed, then how do they compare with the experimental ones ? And could you please show them as Supporting Information?

Response: We first thank this reviewer for identifying our work as a significant progress in the field. We fully accept the request of this reviewer for providing additional details on the computational work reported on in our paper. We have subjected the computational method section to a major revision and now provide a much more detailed presentation of our "crystal structure prediction", or CSP, methodology. Some additional information has also been added to the supporting information.

The “CSP” approach is briefly described here: The ditopic cNDI-linkers, irrespective of the size of attached “steric control units”, form isorecticular Zn-SURMOF-2 structures, which have been reported in previous works. In these previous works, also DFT-calculations where the whole SURMOFs were optimized have been described (DOI: 10.1038/srep00921; DOI: 10.1002/chem.201702968). With this previous information in hand, we have assembled the (geometry-optimized) ditopic linkers into a SURMOF-2 structure using a MOF constructor (DOI: 10.1021/jp507643v; Ref. 27 in the manuscript). Test experiments have shown that the unit cell dimensions along the [100]-direction and the [001] direction do not change upon substitution the cNDI-linkers used here with R-groups, so the same dimensions were used in all calculations. The unit cell length along the [010]-direction has been determined by calculations for a dimer.

This is much clearer to me and should be incorporated in the manuscript main text or Supporting Information. However,

- DOI: 10.1038/srep00921 reports semiempirical UFF and DFTB calculations
- DOI: 10.1002/chem.201702968 is purely experimental

Therefore I don't understand why you quote these two references when stating that “DFT-calculations where the whole SURMOFs were optimized have been described” (lines 302-303 in the Computational methods section).

A more detailed description of these calculations has been added to the method section of the revised manuscript. We believe that the present version of the “CSP” methodology is comprehensive and answers all the queries of the reviewer.

A real effort has been made throughout the Computational methods section but some points remain to be corrected and are detailed right below and later in this document

- Line 299, Computational details: “to calculate Coulomb coupling by the approach of Howard *et al.*^[49]” This is still unclear as Howard reports and compares several approaches. Please be more specific.
- Line 319, the convergence criterium on the force was said to be 10 kJ/mol/nm and is now said to be 0.001 kJ/mol/nm. Why?
- Lines 341-345: I don't understand these sentences and I don't think they should be in this section anyway. “We demonstrated the change in absorption spectra (which is connected to the Coulomb coupling between the neighboring cNDI cores of SURMOFs) obtained within the scope of TD DFT level of theory. To demonstrate that we simulated the absorption properties of dimers for three angles, for which we believed that the difference between the J- and H-aggregates can be easily demonstrated.”

We would like to stress that the message of the paper is that this approximate method is indeed helpful to find a functional group R for which the initially dark SURMOF is turned into a highly emissive one. The goal of the paper is not to provide a through theoretical description of all the different SURMOF structures considered.

OK.

More detailed comments and questions can be found below, in order of appearance. There are quite a number of them, showing how unclear this manuscript can be. The most important ones call the line number in bold font.

To conclude, the concept and the computational strategy seem promising, but I am not yet convinced that the latter is fully valid. Nevertheless I am willing to reconsider a revised manuscript following major revision involving, above all, a real effort of clarification on what has been computed, why and how.

Response: Again, we accept the criticism of the referee and have considered every single critical item in his/her list. Clearly, these informations are crucial to the concept presented here and adding a more detailed discussion has substantially improved the quality of our manuscript. In the revised manuscript, we have included every important detail of the computational methods. Below, we provide answers to all questions raised in a point-by-point fashion.

- Lines 63-66, 'This general interest has stimulated a large number of efforts to enforce J-type aggregation of chromophores into supramolecular assemblies.[11-13] However, very often the resulting aggregates do not have the desired properties, instead of high luminescence, nonradiative quenching processes result in “dark” materials.[4]'

Quoting references 11-13 is perfectly appropriate here but it is not correct to link them up with dark materials, since the materials reported in refs 11-13 are not dark.

Response: We do not understand this criticism. In fact, we did not link Ref. 11-13 to dark materials. Only Ref. [4] is linked to such non-luminescent assemblies. Therefore, we feel that no changes are required in this case.

They are linked because they are sequential, but you can leave it as it is.

- Lines 72-78, 'A typical example is the case of 9-anthracene carboxylic acid (ACA), a small, prototype chromophore. Non-substituted ACA crystallizes in a triclinic structure, with bright photoluminescent (PL) properties. When side groups are attached at the 10- position of ACA to modify the position of emission bands, the slightly modified chromophores crystallize in rather different crystal structures (triclinic, monoclinic, and orthorhombic).[17] As CSP methods struggle to predict the experimentally realized structures, researchers had to resort to trial-and-error strategies to optimize the photophysical properties of such ACA assemblies.'

Stressing on CSP seems misleading to me since this work does not report improved CSP approaches. Instead I would suggest to add something like '[serendipitous discovery] or extensive systematic screening[17]' at the end of line 71.

Response: We do not agree with the referee. Indeed, what we report is the prediction of crystal structures by using an approximate scheme as described above. Reducing the number of degrees of freedom to essentially one (rotation of the cNDI around linker axis) makes CSP much easier, and this is why our approach was successful. This is a central point of the paper, and we would like to stay with the original wording.

With the additional computational details given in the revised version, and with the additional explanations given in this response, the situation is clearer but not everywhere, unfortunately. I shall return to this point later.

- Line 104 'rather close packing'

Ref 39 says 5.4 angstroms, could you please specify this number since it is used later as a reference?

Response: Actually, the value 5.4 Å relates to a SURMOF with a different cNDI-linker. The cited paper (ref 39) demonstrated that for inter-sheet distance of ~ 5.4 Å strong excitonic couplings are present among the chromophores. Please note, that for the NDI(OEt)₂ linker used

in the present SURMOFs with R=H the inter-chromophore distance is 5.8 Å, and this has been clearly mentioned in the manuscript.

Thank you for clarifying this point. However I found this information in line 147 but not in line 105, while the latter has been highlighted in yellow in the revised version.

- Line 113 'for theta > 55.4°

Could you please comment on this number with respect to the 54.7° magic angle (e.g. as in ref. 5)?

Response: The magic angle of 54.7° is obtained when using a simple dipole-dipole approximation. In our calculations, we used the more sophisticated TrEsp approach, where transition charges were fitted to every atom in our monomer from electrostatic potential fit (as in reference 43).

If so, my advice is to add this comment because it highlights your method.

In fact, the monomer used to calculate the transition electrostatic potential is modified in the DFT optimization of the organic linkers, and the final structure is not completely planar, (see last part of SI where all coordinates and partition charges used in Coulomb coupling calculations are given).

Do you expect me to build the structures from the coordinates given in SI, and then compare myself with the few parameters given in the main manuscript, in order to assess the differences between DFT-optimized and MD-optimized geometries? I shall return to this point later.

As a result, the final angle is slightly different from the magic angle obtained from simpler models.

If so, my advice is to add this comment because it highlights your method.

- Line 113 'the Coulomb coupling'

The signs of the Coulomb coupling JC and the charge-transfer coupling JCT can vary independently.

Could you please comment on the possible interference between Coulomb and CT couplings (e.g. J. Chem. Phys. 2015, 143, 244707) ? Why do you think JC is sufficient to predict the type of aggregate ? In other words, why would Kasha's seminal exciton model hold here ?

Response: Charge-transfer coupling (JCT) is usually present within “tightly stacked” organic systems with small (below 4 Å) intermolecular distances. Since in our case the minimal distance between the cNDI cores is much larger, namely 5.8 Å for R=H, and increases up to 6.8 Å in the case of the R=iPr, we conclude that the charge-transfer couplings will be small, in fact much smaller than the Coulomb coupling between the neighboring chromophores. This expectation is confirmed by the inspection of HOMO and LUMO orbital overlaps, which are small. Altogether these observations give us confidence that we can base our theoretical explanation only on dominating Coulomb coupling.

I fully agree. My advice is to add a short comment in order to strengthen your argument.

Line 119 'assemble MOFs in silico'

This is not consistent with the computational methods

Response: As already stated above, we used a MOF constructor developed by some of us earlier (DOI: 10.1021/jp507643v; Ref. 27 in the manuscript) to take ditopic linkers and construct a SURMOF-2 structure. We believe that for this process the wording “assemble MOFs in silico” is appropriate. We have added an additional reference to make this point more clear.

- The article corresponding to the cited DOI is « AuToGraFS: Automatic Topological Generator for Framework Structures » by Matthew A. Addicoat,* Damien E. Coupry, and Thomas Heine. There are no common authors with the current manuscript, so what does “some of us” mean?
- Do you expect me to compare the two lists of references to find out what the additional reference is?
- Dimers and trimers placed at given distances and orientations, as they would be in SURMOFs, are certainly very pertinent models. But I maintain that lines 118-121 are not correct: « Therefore, we first created a library of possible SCUs and then used simulation schemes to assemble MOFs *in silico* from the resulting 18 linkers,^[27] optimizing their geometry, and determining the resulting rotation angles θ for each SCU. The corresponding results (Figure 1b) [...] .» These sentences mean that the data reported in Figure 1b was obtained from MOF *in silico* structures, which is simply wrong because it was obtained from Gromacs-optimized model trimers.
- Lines 145-148 are also incorrect: “A combination of molecular dynamics (MD) based simulation and density functional theory (DFT) predicts a NDI(OEt)₂ geometry in the Zn-(NDI(OEt)₂) (A) SURMOF with a co-facial stacking (~5.8 Å) of the NDI(OEt)₂ cores, leading to an undesirable H-aggregate type arrangement (See method section and Supporting Figure 2).” Supporting Figure 2 shows a DFT-optimized trimer but neither MD-optimized geometry nor SURMOF.

Line 200

Is it correct to say that Supporting Figure 5 illustrates the change in dihedral angle between R=H and R=iPr? If so, please replace ‘rotational angle’ by ‘dihedral angle’ in line 200, and reconsider the caption of Supporting Figure 5, for consistency with the words ‘dihedral angle’ and ‘rotation angle’ as defined in Supporting Table 1.

- Figure 1

Please keep the same colour code throughout.

Fig 1a : Ph are blue, zinc is green ; Fig 1b : Ph are green

Response: We have modified Fig. 1 – now the color scheme should be consistent.

Good.

Fig 1a : R groups are red ; Fig 2a : R groups are blue

Response: These problems have been removed.

Good.

- Figure 1b

* 'TDM-TDM coupling': please expand TDM acronym once (or in text line 140)

Response: We have added the expanded acronym in text line 140.

Good.

* caption 'Coulomb coupling vs rotation angle theta (as shown in Figure 1a at the right hand side)': theta is in fact shown in the inset of Figure 1b

Response: This problem has been removed.

Good.

* what does the green line correspond to ?

Response: The green line corresponds to the calculated TDM-TDM couplings for different rotation of NDI(OEt)₂, as mentioned in the figure caption.

If I understand correctly (after seeing the additional part of Supporting Figure 3), the line is the coupling, and you have added a few discrete marks to show examples for a few specific R groups.

Line 128 'Crystal structure prediction of NDI(OEt₂)-assembly'

Again if the calculations are not made on SURMOF structures, this title is misleading. Something like 'Structural prediction of a library of NDI assemblies' would be more correct, without diminishing the quality or the impact of the work.

Response: As mentioned before, we have used a MOF-constructor to assemble a SURMOF-2 from the ditopic linkers and then used MD-simulations to optimize the unit cell size along the [010] crystallographic direction. We agree that this is not a full structure determination. Following the suggestion of the reviewer, we have revised the line 128 as follows: "Structural prediction of a library of NDI(OEt)₂ assemblies".

Thank you for clarifying this point.

- Line 135 'co-facial stacking of the cores'

Please report the interchromophoric distance (in the text or on Supporting Figure 2)

Response: The missing information has been added to text.

Good.

- Line 141 'closely spaced chromophores'

Please report the interchromophoric distance

Response: The missing information has been added to text.

Good.

- Line 143 'placed at a distance (x) of 6.8 angstroms'

Is that what 'closely spaced' means ? This is significantly larger than the 5.4 angstroms distance of ref 39.

Where does this number come from ?

Response: This number has been obtained by determining the [010] unit cell parameter from the MD calculations for a non-rotated (R=H) cNDI linker. This value is larger than the 5.4 Å reported previously since the linkers are quite different..

Fair enough (Gromacs step ii). So why don't you write this first sentence somewhere, for clarity?

Line 145 'for theta > 55.4°'

Where does this number come from? Isn't there a plot missing ? Second half of Supporting Figure 3 'angle dependence of monomers', J_{mn} vs theta ?

Response: This number comes from Figure 1b where the angle dependence of the Coulomb coupling for a fixed interchromophoric distance of 6.8 Å is displayed (green line). The same plot is now also introduced in the Supporting Figure 3.

Good.

- Line 158 'apparently more bulky groups yielded a smaller angle of theta'

In the methyl/benzyl comparison, this sentence should mention that they both induce similar dihedral angles (according to Supporting Table 1). Otherwise the comparison does not hold.

Response: This problem has been fixed.

- Does this correspond to the modifications in line 122 and lines 169-170 ? If so, then I disagree the problem is not fixed. Steric bulk induces changes both in dihedral and rotation (theta) angles. The comparison on rotation angles only does not hold without mentioning that the dihedral angles are similar.
- Besides I would have thought that 34° and 41° were significantly different, whereas they are now quoted as 'similar'. Why?

- Line 158 'methyl (9)'

In Figure 2a, methyl is 8

Response: This problem has been fixed.

Good.

Line 159 'benzyl (2)'

In Figure 2a, benzyl is 6

Response: This problem has been fixed.

Good.

- Line 172: “we chose three R groups, two from the borderline region in Figure 1. The borderline region is well defined on Figure 2b (blue square), but how is it defined in Figure 1 ?”

- Line 167, figure 2b

I understand that the rotation angles were obtained from Gromacs-optimized trimers, is this correct ? If so, this should be clarified.

Response: Yes, those were obtained from GROMACS. We have augmented the text of the figure caption to make this point clear.

Good but please make it completely clear, the caption should be able to stand alone without constantly requiring to jump to the computational details. It's only a matter of adding the word 'trimers'.

- Line 195 'Supporting Information Figure 5' is in fact Supporting Figure 6

Response: The supporting figures have been revised accordingly.

Good.

- Line 204 'Supporting Information Figure 6' is in fact Supporting Figure 5

Response: The supporting figures have been revised accordingly.

Good.

Line 207 'vibronic overtones at 436 and 472 nm'

Does the vibrational spacing fit with a specific vibrational mode? Which one? Does it also fit with the simulated spectra shown in Supporting Figure 7?

Response: Yes, the vibronic features in the solvated cNDI-units can be related to vibrational modes, as has been discussed previously in the literature (DOI: 10.1039/c0cc00078g). However, this is not relevant for our work, since the vibronic features changed in the corresponding SURMOF data (as a result of strong excitonic coupling, see DOI:10.1002/anie.201708267). Since vibronic couplings were not included in the simulations, a direct comparison is not meaningful here. Please note also, that the absorption bands are shifted in comparison to

experimental values, evidently due to the presence of solvation effects in the experimental results for the solvated monomers.

My suggestion was indeed to include vibronic coupling in the UV-vis simulation of monomer and dimers (Supporting Figure 7). This calculation could have produced a valuable addition.

Line 212 'We noted that the coupling energy for A and 9A or 15A did not differ markedly (Figure 1b, ~50 meV)'

The data for R=iPr (as in SURMOF 15A) is shown on Figure 1b and is slightly negative (roughly -20meV) ; the data for R=Et (as in SURMOF 9A) is shown on Figure 1b and is slightly positive (roughly 20 meV) ; the data for R=H (as in SURMOF A) is not shown on Figure 1b. This doesn't make sense.

Response: The referee is correct. We have included the predicted θ for R=H by GROMAC-2018.4 in Figure 1b.

Thank you for adding the mark for R=H.

However I still don't understand why you are saying that the values are similar and close to 40 meV (line 219). To me they are neither similar, nor close to 40 meV.

The values I can read on Figure 1b are about +20 meV for R=Et, and about -20 meV for R=iPr. So please correct lines 218-219 accordingly, or explain again why your sentence is correct as it is.

Those numbers can also be seen more easily on Figure 4d, which, by the way, has been replotted.

- previously plotted between -12 and +32 meV, with three data points estimated at -8; +18; +30;
- now plotted between -20 and +25 meV, with three data points estimated at -16; +21; +23.

Why and how did you replot this figure?

Line 216 'To explain this behavior we employed time dependent density functional theory (TD-DFT) to simulate the electronic absorption spectra of the NDI(OEt)₂ dimers with rotation angles 34 and 60° for H and J-type coupling, respectively (Supporting Information Figure 7 and 8)'

* How were these angles chosen ? The angles reported in Supporting Table 1 for R=Me or Et and R=iPr could have been chosen advantageously. Also see the vague expression 'certain angle' line 317.

Response: We are grateful for this comment by the referee. Briefly, we took two rotation angles (34 and 60 °), which clearly represent different aggregation types, H and J, respectively. A corresponding statement has been added to the manuscript.

Good

* What was the interchromophoric distance in these TD-DFT calculations? Also see the vague expression 'proper distance' line 316.

Response: We have added the requested information (6.8 Å) to the manuscript.

Good

* What is the origin of the additional red shoulder in the computed absorption spectrum of the 60°-tilted dimer ? If this shoulder is of vibronic origin, a vibrationally-resolved spectrum should be computed to confirm it. If, on the other hand, this shoulder is due to an additional electronic transition, it would be interesting to identify it.

Response: The origin of the additional transition at 485.62 nm in the absorption spectrum is an electronic transition. The known fact is that due to the parallel transition dipole orientations within coordinated chromophores in close spatial proximity low lying transitions are forbidden in the case of non-emissive H-aggregates. However, when two chromophores form a J-aggregate and two transition dipoles become almost collinear, this dark state becomes allowed and low lying visible absorption peaks are observed in the spectrum (DOI: 10.1039/C0DT01226B). From the comparison of the absorption spectra for the case of 34° and 60° in the Supporting Figure 8 (c and d), there is a visible rise in intensity of the previously dark singlet transition S1. From the analysis of the leading contributions for S1 transition it can be seen that it is dominated by HOMO-LUMO transition localized on central part of both chromophores.

Thank you for clarifying this point, I fully agree. The revised Supporting Figure 8 b) c) d) now allows the reader to compare the UV-vis absorption spectra for three different theta values, which is highly illustrative. Please add this explanation in the main text (lines 225-227) instead of the current very poor comment : “But, in J-type a new absorption shoulder was observed in the longer wavelength region (~485 nm).”

Line 223

Why are 8A and 9A so different in absorption (Figure 4a) but so similar in emission (Figure 4b,c) and structure (Supporting Table 1) ?

Response: 8A and 9A absorption spectra are different because their transition dipole moment (TDM) couplings are different (in line with the theoretical predictions).

I disagree with this. The computed theta values for R=Me and R=Et are very similar (40.9 vs 42.5°, Supporting Table 1) so the TDM-TDM couplings should be very similar too.

Although the steady state emission spectra (Figure 4b) of 8A and 9A do not differ significantly, there is an evident difference in PLQY (Figure 4d) for these two species. A more detailed answers to this questions would require a thorough theoretical analysis, which we feel is beyond the scope of the present paper. In fact, the message of the paper is that we use computational screening to find side groups which turn the “dark” SURMOF into an emissive one. The success of this approach is evident from the experimental observation that the SURMOF fabricated from the optimized linkers indeed is highly emissive.

OK.

- Figure 4a

The colour code seems wrong with an inversion between 8A and 9A (inconsistency with the text and with the caption of Figure 3)

Response: We are grateful to the referee for pointing out this problem. Figure 3 has been corrected and the color codes are now consistent.

Good.

- Figure 4b

The partly hidden caption should be completely hidden

Response: Nothing is hidden in Figure 4b, it is a relevant legend.

My mistake, sorry. I got confused between 15 and 15A.

Figure 4d

* Please provide the emission spectra corresponding to Figure 4d-inset(ii) as Supporting Information

Response: We have followed the suggestion of the referee and have added the requested material to the supporting information.

- Do you mean the new Supporting Figure 10, that was previously shown as an inset of Figure 4d? I suggested you to show the emission spectra, in addition to the photos. The PL spectrum of 15 is indeed shown on Figure 4b, but the PL spectrum of the linker with R=H is not shown.
- Besides, the caption of Supporting Figure 10 is incomplete with respect to the caption of the previous Figure 4d : “SURMOF thin films deposited on a quartz substrate” has become “SURMOFs”.

* are the TDM-TDM coupling values shown on Figure 4d supposed to match some of those shown on Figure 1b ? It is not clear to me what the difference between these two sets of data is.

Response: We thank the reviewer for pointing out this issue. We have chosen the TDM-TDM coupling values from figure 1b, which were computed for the initially predicted rotation angles of 8A, 9A and 15A.

Do you mean that the initial Figure 4d used other TDM-TDM values because they used other rotation angles? Angles from MD-optimized trimers, instead of angles from DFT-optimized trimers? Or yet other angles?

Figure 4d shows a correlation of prediction and experimental results. In the revised figure 4d, the values can be correlated perfectly with figure 1b and supporting table 1. We have also mentioned in the figure caption of figure 4d, that the TDM-TDM couplings plotted are from Figure 1b.

OK

Line 264 'crystal structure prediction'

Same remark as above : at this stage I can't say if and how any SURMOF structure was computed at all.

Response: As explained above we feel that using this term is justified. We have used approximate methods to predict SURMOF structures, we did not claim that the SURMOF structure was determined using a full DFT structure optimization.

My concern is that you are showing monomers, dimers, trimers, but you are not providing any SURMOF structure, unlike DOI: 10.1038/srep00921.

- Figure 1b, Figure 2b, Figure 4d, Supporting Table 1, Supporting Figure 2, Supporting Table 2, Supporting Figure 5 : concern trimers
- Supporting Figure 3, Supporting Figure 7, Supporting Figure 8, Supporting Table 3 : concern dimers

I'm not asking at all for a full DFT structure optimization. I'm only asking for clarity, consistency and rigour.

Line 267 '+40 meV'

* Was this computed for a dimer with R=H or for SURMOF A?

Response: This was computed for a dimer with R=H.

- OK so please specify it, because at the point where this is written, it seems you are talking about data for the SURMOFs themselves.
- In addition, why did you change the +40 meV value into +22 meV?
- And how do you get 'almost a factor of two' between 22 and 16?

* Similarly, '-15 meV', was this computed for a dimer with R=iPr or for SURMOF 15A?

Response: This was computed for a dimer with R=iPr.

Same remark as above, please specify it.

Line 293 'Optimization of the layer distance via calculating the potential energy of neighboring linkers as a function of layer spacing (5.0~8.0 angstroms)'

Is this performed on dimers or on MOF structures? Since step (i) concerns monomers and step (iii) concerns trimers, I would rather think that step (ii) concerns dimers, in which case there are no calculations on SURMOF structures.

Response: This optimization was performed on a dimer of cNDI with related SCU attached to it. Thank you for clarifying this point.

- Line 307 'Taking the optimized potential energy value as criteria, the geometry obtained via SA-MD is substantially better than without SA-MD optimization'

Please give elements of comparison (distances, energies).

Response: Following is a comparison of potential energy values obtained with and without SA-MD: By Using SA-MD optimization, for linker **13**, the monomer (step i) potential energy value decreased by 32 kJ/mol, and for trimer (step iii) it is 53 kJ/mol.

OK

- Line 316 'proper distance'

For clarity, please recall these numbers for each R group.

Response: The requested information has been added to the manuscript.

Thank you for clarifying this point.

- Line 317 'certain angle'

For clarity, please recall these numbers for each R group.

Response: The requested information has been added to the manuscript.

Thank you for clarifying this point.

- Line 326 'Becke D3 dispersion'
should be 'Grimme D3 dispersion'.

Response: We have changed the wording in accord with this suggestion.

Good.

- Line 327 'high quality grid'

Please be more specific. Also please specify which program was used for the DFT and TD-DFT calculations.

Response: We have added the requested information to the paper.

Good

- Line 391 'photochemical' should be 'photomechanical'.

Response: Has been corrected.

Good.

- Supporting Information Table 1

If I understood correctly, these calculations were performed on Gromacs-optimized trimers. For clarity this information could be recalled in the Table caption.

Response: We have added the requested information to the manuscript.

Good but please make it clear as well in the Supporting Table 1 caption, which should be able to stand alone. It's only a matter of adding the word 'trimers'.

In fact the rotation angles in Supp. Table 1 are the ones that are plotted on Figure 2b, so please indicate this link in the caption of Supp. Table 1 too, in order to help the reader.

Supporting Information Figure 2

* Please specify interchromophoric distance

* If the calculation indeed concerns a SURMOF structure, how was it performed and where are the zinc ions ?

Response: The values for the inter-chromophore distance have been added to the manuscript. The calculations were performed for a trimer model, without considering the zinc paddle-wheel units. This information has been added to the legend of the figure.

The caption now seems correct (the initial one wasn't), and consistent with the computational methods.

Supporting Information Figure 3

The angle dependence of the Coulomb coupling should be shown.

Response: Please note that the angle dependence is already present Figure 1b of the original manuscript. In order to account for the criticism of the referee, we have added this information to Figure 3 (bottom).

My question was related to the fact that Supp. Figure 3 seemed incomplete as it indicated "angle dependence of monomers" without any graphical support (cf red oval that I added on my screen capture).

140

141 Supporting Information Figure 3: Coulomb coupling dependence on distance: Calculated

142 Coulomb coupling between two NDI(OEt)₂ monomers: distance dependence for rotation angle

143 0°.

144

145

146

147

Supporting Information line 151 'The shift in (010) diffraction from A to the other SURMOFs indicate a slight changes (~0.1 angstrom) in the inter-sheet distances'

Computed intermolecular distances given in Supporting Table 1 (presumably in trimers, not in SURMOFs) show that the intermolecular distances vary by 1 angstrom (from 5.8 angstroms for R=H to 6.8 angstroms for R=iPr). Isn't this a concern to you that this doesn't fit with the experimental intermolecular distances, which are the same for all SURMOFs?

Response: We are grateful to the referee for pointing out this problem. The sentence has been revised. From A to 15A, the change in inter-sheet distance is ~1 Å, but not ~0.1 Å (also mentioned in the main text). So the calculation and experimental values agree well.

Good to know!

- Supporting Information line 155

For clarity please specify 'distances as in respective experimental SURMOF structures'.

Response: We have modified the manuscript in accord with this suggestion.

Please recall these distances in Supporting Table 2 as well (e.g. additional column). It's really irritating to have to chase numbers all the time!

Supporting Information Table 2

How do you explain that the angles in Supp. Table 1 (from Gromacs-optimized trimers) are so different from the angles in Supp. Table 2 (from DFT-optimized trimers)? 24-28° instead of 41-42°, and 53° instead of 63°.

Given the angular variation of the TDM-TDM coupling, it is crucial to have reliable angles, especially in the region of the magic angle.

Please rephrase lines 197-198 accordingly: “The DFT simulated linker orientations in all four structures predict different rotation angles (Supporting Information Table 2).”

Supporting Information Figure 6

* The caption says 'DFT simulated geometries of A and 15A', which is very confusing with respect to the Computational methods section. Is there any zinc in these calculations? It seems as if these calculations were performed on monomers.

Response: We have modified the manuscript in accord with this suggestion..

The caption now seems correct (the initial one wasn't), and consistent with the computational methods.

* According to Supporting Table 1, the dihedral angle between the NDI core and the Ph substituents is 69.1° for R=H and 87.4° for R=iPr. This is not obvious from Supporting Figure 6.

Response: We have modified the manuscript in accord with this suggestion.

Good.

- Supporting Information line 172

* Please specify interchromophoric distance in the dimers.

Response: We have modified the manuscript in accord with this suggestion. In the revised version these distances are mentioned.

Good.

* Were they optimized as in Gromacs step (ii) ?

Response: Yes, we obtained this from GROMACS-2018.4 based calculations (in step ii).

OK

- Supporting Information Figure 8

* Please specify interchromophoric distance in the dimer.

Response: Now the distances are mentioned.

Good.

* Please show 60° angle on the figure. Is this the theta angle of Figure 1b?

Response: Now we have added this in the revised version.

Good.

Unfortunately, we do not understand the question “Is this the theta angle of Figure 1b?”. Figure 1b and SI Figure 8 are clearly described in the corresponding figure captions.

OK

* Is the image at the left of the computed absorption spectrum a sort of top view of the one shown in the inset of the spectrum ?

Response: Yes, it is the top view. It is described in the revised version.

Good.

* Please show the full spectrum (the red region is currently cut)

Response: The red parts of the spectra were deliberately omitted due to the fact that there are no electronic transitions. The additional Table 3 in the Supporting Information is now added, with the overview of the all calculated transitions and corresponding wavelengths. This clearly shows that lowest singlet transitions for all chromophore assemblies lay bellow 500 nm.

I agree that there are no additional peaks above 500 nm. However sometimes tails or shoulders (or absence of) are important to show.

* 'top' and 'bottom' do not correspond to the current display

Response: We have modified the manuscript in accord with this suggestion.

Good.

Point-by-point response

A number of red lights have turned yellow or green with this revised version, but a few red lights remain. Please reconsider the points written in red (major) and highlighted in yellow (minor). I know there are strong space constraints, but there are ways to ensure that the space constraints aren't detrimental to science.

Response: We have responded to all the concerns (red and yellow) raised by the reviewer, and made necessary changes to the manuscript. All the changes in the manuscript are highlighted in yellow, and the responses are written in blue.

Reviewer #3 (Remarks to the Author):

This manuscript by Wöll et al. reports a joint theoretical and experimental study on luminescent MOFs. As a continued effort in the framework of an established KIT-CEISAM collaboration, the current work uses a MOF structure to assemble naphthalenediimide (NDI) chromophores in such a manner as to obtain luminescent materials. Using steric variations to control aggregation properties is certainly a classical but efficient approach. The attractive side of this work lies in the computational screening of a library of substituents, ahead of synthesis, to guide the design of MOFs displaying promising luminescence properties. As such, this work represents a significant progress in the field. However I have a major concern precluding acceptance of this manuscript, regarding the technical aspects of the theoretical contribution. The undertaken computational strategy is indeed appealing, but given the data reported in the main text and as Supporting Information, I am currently unable to say if the strategy is valid. In principle the 'Computational methods' section is meant to enable the reader to reproduce the calculations. With the given level of details, I doubt anyone could do so. Such a lack of rigour is deleterious, and a tremendous effort should be put in clarifying this. First and foremost, the only mention of calculations on a SURMOF structure (which would be required to claim reporting crystal structure prediction, CSP) is a sentence at the beginning of the article (lines 134-137 : 'Density functional theory (DFT) predicts a NDI(OEt)₂ geometry in the Zn-(NDI(OEt)₂) (A) SURMOF with a co-facial stacking of the NDI(OEt)₂ cores, leading to an undesirable H-aggregate type arrangement (See method section and Supporting Figure 2)'. However the method section says nothing about this calculation, and Supporting Figure 2 shows a trimer but no zinc. Elsewhere, only monomers, dimers or trimers of NDI are mentioned. To mimic SURMOF structures, computations on dimers or trimers with appropriate interchromophoric distances and tilt angles (these numbers being scarcely mentioned) could well be meaningful, but my gut feeling is that the confusion on what has exactly been computed sheds doubt on this validity. I am perfectly

happy with using clever and sensible models, I just need to be convinced that the models are indeed sensible, and how they were computed. If I misunderstood what has actually been computed, i.e. if SURMOF structures 8A, 9A and 15A were indeed computed, then how do they compare with the experimental ones ? And could you please show them as Supporting Information?

Response: We first thank this reviewer for identifying our work as a significant progress in the field. We fully accept the request of this reviewer for providing additional details on the computational work reported on in our paper. We have subjected the computational method section to a major revision and now provide a much more detailed presentation of our “crystal structure prediction”, or CSP, methodology. Some additional information has also been added to the supporting information.

The “CSP” approach is briefly described here: The ditopic cNDI-linkers, irrespective of the size of attached “steric control units”, form isoreticular Zn-SURMOF-2 structures, which have been reported in previous works. In these previous works, also DFT-calculations where the whole SURMOFs were optimized have been described (**DOI: 10.1038/srep00921; DOI: 10.1002/chem.201702968**). With this previous information in hand, we have assembled the (geometry-optimized) ditopic linkers into a SURMOF-2 structure using a MOF constructor (**DOI: 10.1021/jp507643v**; Ref. 27 in the manuscript). Test experiments have shown that the unit cell dimensions along the [100]-direction and the [001] direction do not change upon substitution the cNDI-linkers used here with R-groups, so the same dimensions were used in all calculations. The unit cell length along the [010]-direction has been determined by calculations for a dimer.

This is much clearer to me and should be incorporated in the manuscript main text or Supporting Information. However, • **DOI: 10.1038/srep00921** reports semiempirical UFF and DFTB calculations • **DOI: 10.1002/chem.201702968** is purely experimental Therefore I don't understand why you quote these two references when stating that “DFTcalculations where the whole SURMOFs were optimized have been described” (lines 302-303 in the Computational methods section).

Response: We are pleased to learn that the referee likes this new text. It has been added to the method section of the present manuscript.

According to the suggestion, we have removed the reference (**DOI: 10.1038/srep00921**).

A more detailed description of these calculations has been added to the method section of the revised manuscript. We believe that the present version of the “CSP” methodology is comprehensive and answers all the queries of the reviewer.

A real effort has been made throughout the Computational methods section but some points remain to be corrected and are detailed right below and later in this document

- Line 299, Computational details: “to calculate Coulomb coupling by the approach of Howard *et al.*[49]” This is still unclear as Howard reports and compares several approaches. Please be more specific.

Response: In the revised version, we have expanded the sentence and explicitly stated that we used the sum over interaction between atomic transition charges. In the paper by Howard *et al.* this approach has been shown to be suited best.

- Line 319, the convergence criterium on the force was said to be 10 kJ/mol/nm and is now said to be 0.001 kJ/mol/nm. Why?

Response: We are grateful to the referee for pointing out this inconsistency. In all cases, the convergence criterium on the force was 0.001kJ/mol/nm. The corresponding corrections have been made to the manuscript.

- Lines 341-345: I don't understand these sentences and I don't think they should be in this section anyway. “We demonstrated the change in absorption spectra (which is connected to the Coulomb coupling between the neighboring cNDI cores of SURMOFs) obtained within the scope of TD DFT level of theory. To demonstrate that we simulated the absorption properties of dimers for three angles, for which we believed that the difference between the J- and H-aggregates can be easily demonstrated.”

Response: We have reworded this text in order to account for the criticism of the referee. It now reads: “Calculations carried out at the TD DFT level of theory also yielded a similar change in absorption spectra. They can be attributed to the Coulomb coupling between the neighboring cNDI cores of SURMOFs.”

We would like to stress that the message of the paper is that this approximate method is indeed helpful to find a functional group R for which the initially dark SURMOF is turned into a highly emissive one. The goal of the paper is not to provide a thorough theoretical description of all the different SURMOF structures considered.

OK.

More detailed comments and questions can be found below, in order of appearance. There are quite a number of them, showing how unclear this manuscript can be. The most important ones call the line number in bold font.

To conclude, the concept and the computational strategy seem promising, but I am not yet convinced that the latter is fully valid. Nevertheless I am willing to reconsider a revised manuscript following major revision involving, above all, a real effort of clarification on what has been computed, why and how.

Response: Again, we accept the criticism of the referee and have considered every single critical item in his/her list. Clearly, these informations are crucial to the concept presented here and adding a more detailed discussion has substantially improved the quality of our manuscript. In the revised manuscript, we have included every important detail of the computational methods. Below, we provide answers to all questions raised in a point-by-point fashion.

- Lines 63-66, 'This general interest has stimulated a large number of efforts to enforce J-type aggregation of chromophores into supramolecular assemblies.[11-13] However, very often the resulting aggregates do not have the desired properties, instead of high luminescence, nonradiative quenching processes result in “dark” materials.[4]'

Quoting references 11-13 is perfectly appropriate here but it is not correct to link them up with dark materials, since the materials reported in refs 11-13 are not dark.

Response: We do not understand this criticism. In fact, we did not link Ref. 11-13 to dark materials. Only Ref. [4] is linked to such non-luminescent assemblies. Therefore, we feel that no changes are required in this case.

They are linked because they are sequential, but you can leave it as it is.

- Lines 72-78, 'A typical example is the case of 9-anthracene carboxylic acid (ACA), a small, prototype chromophore. Non-substituted ACA crystallizes in a triclinic structure, with bright photoluminescent (PL) properties. When side groups are attached at the 10- position of ACA to modify the position of emission bands, the slightly modified chromophores crystallize in rather different crystal structures (triclinic, monoclinic, and orthorhombic).[17] As CSP methods

struggle to predict the experimentally realized structures, researchers had to resort to trial-and-error strategies to optimize the photophysical properties of such ACA assemblies.'

Stressing on CSP seems misleading to me since this work does not report improved CSP approaches. Instead I would suggest to add something like '[serendipitous discovery] or extensive systematic screening[17]' at the end of line 71.

Response: We do not agree with the referee. Indeed, what we report is the prediction of crystal structures by using an approximate scheme as described above. Reducing the number of degrees of freedom to essentially one (rotation of the cNDI around linker axis) makes CSP much easier, and this is why our approach was successful. This is a central point of the paper, and we would like to stay with the original wording.

With the additional computational details given in the revised version, and with the additional explanations given in this response, the situation is clearer but not everywhere, unfortunately. I shall return to this point later.

- Line 104 'rather close packing'

Ref 39 says 5.4 angstroms, could you please specify this number since it is used later as a reference?

Response: Actually, the value 5.4 Å relates to a SURMOF with a different cNDI-linker. The cited paper (ref 39) demonstrated that for inter-sheet distance of ~ 5.4 Å strong excitonic couplings are present among the chromophores. Please note, that for the NDI(OEt)₂ linker used in the present SURMOFs with R=H the inter-chromophore distance is 5.8 Å, and this has been clearly mentioned in the manuscript.

Thank you for clarifying this point. However I found this information in line 147 but not in line 105, while the latter has been highlighted in yellow in the revised version.

- Line 113 'for theta > 55.4°'

Could you please comment on this number with respect to the 54.7° magic angle (e.g. as in ref. 5)?

Response: The magic angle of 54.7° is obtained when using a simple dipole-dipole approximation. In our calculations, we used the more sophisticated TrEsp approach, where transition charges were fitted to every atom in our monomer from electrostatic potential fit (as in reference 43).

If so, my advice is to add this comment because it highlights your method.

Response: In following the suggestion of the referee, we have added such a sentence to the method section.

In fact, the monomer used to calculate the transition electrostatic potential is modified in the DFT optimization of the organic linkers, and the final structure is not completely planar, (see last part of SI where all coordinates and partition charges used in Coulomb coupling calculations are given).

Do you expect me to build the structures from the coordinates given in SI, and then compare myself with the few parameters given in the main manuscript, in order to assess the differences between DFT-optimized and MD-optimized geometries? I shall return to this point later.

Response: Following the reviewer's request, we have added a view of the linker geometries obtained in (a) MD and (b) DFT which allow seeing the differences in geometry (See appended below and also added to the supporting information Figure 12).

Figure R1: View of the linker (15) geometries obtained by (a) MD and (b) DFT.

As a result, the final angle is slightly different from the magic angle obtained from simpler models.

If so, my advice is to add this comment because it highlights your method.

Response: In following the suggestion of the referee, we have added a sentence to the method section describing this approach.

- Line 113 'the Coulomb coupling'

The signs of the Coulomb coupling JC and the charge-transfer coupling JCT can vary independently.

Could you please comment on the possible interference between Coulomb and CT couplings (e.g. J. Chem. Phys. 2015, 143, 244707) ? Why do you think JC is sufficient to predict the type of aggregate ? In other words, why would Kasha's seminal exciton model hold here ?

Response: Charge-transfer coupling (JCT) is usually present within “tightly stacked” organic systems with small (below 4 Å) intermolecular distances. Since in our case the minimal distance between the cNDI cores is much larger, namely 5.8 Å for R=H, and increases up to 6.8 Å in the case of the R=iPr, we conclude that the charge-transfer couplings will be small, in fact much smaller than the Coulomb coupling between the neighboring chromophores. This expectation is confirmed by the inspection of HOMO and LUMO orbital overlaps, which are small. Altogether these observations give us confidence that we can base our theoretical explanation only on dominating Coulomb coupling.

I fully agree. My advice is to add a short comment in order to strengthen your argument.

Response: Following the reviewers suggestion, we have added a sentence in the related section and highlighted in yellow.

Line 119 'assemble MOFs in silico'

This is not consistent with the computational methods (more details are MOF given below)

Response: As already stated above, we used a MOF constructor developed by some of us earlier (DOI: 10.1021/jp507643v; Ref. 27 in the manuscript) to take ditopic linkers and construct a SURMOF-2 structure. We believe that for this process the wording “assemble MOFs in silico” is appropriate. We have added an additional reference to make this point more clear.

• The article corresponding to the cited DOI is « AuToGraFS: Automatic Topological Generator for Framework Structures » by _Matthew A. Addicoat,* Damien E. Coupry, and Thomas

Heine_. There are no common authors with the current manuscript, so what does “some of us” mean?

Response: We are grateful to the referee for pointing out this inconsistency. We are sorry for this mistake. In the new version of the manuscript, it has been corrected to “by Heine *et al*”.

• Do you expect me to compare the two lists of references to find out what the additional reference is?

Response: The new reference added to explain this point is the paper (DOI: 10.1021/jp507643v), reference number 27 in new manuscript.

• Dimers and trimers placed at given distances and orientations, as they would be in SURMOFs, are certainly very pertinent models. But I maintain that lines 118-121 are not correct: « Therefore, we first created a library of possible SCUs and then used simulation schemes to assemble MOFs *in silico* from the resulting 18 linkers,^[27] optimizing their geometry, and determining the resulting rotation angles θ for each SCU. The corresponding results (Figure 1b) [...] .» These sentences mean that the data reported in Figure 1b was obtained from MOF *in silico* structures, which is simply wrong because it was obtained from Gromacs-optimized model trimers.

Response: In order to account for the criticism of the referee we have modified the text, which now reads: “Therefore, we first created a library of 18 possible SCUs, then optimized the geometry of the individual linkers using a force-field calculation, and then used a simple scheme to assemble 18 different MOFs *in silico* using a previously described MOF constructor.^[27] Then, the MOF lattice constant was fixed and a MD scheme was used to optimize the structure of the linkers (including the intramolecular dihedral angles). As a result of inter-ligand interactions, the dihedral angles are changed and also the rotation angle θ (angle between cNDI core and carboxylate-planes) changed.”

• Lines 145-148 are also incorrect: “A combination of molecular dynamics (MD) based simulation and density functional theory (DFT) predicts a NDI(OEt)₂ geometry in the Zn-(NDI(OEt)₂) (A) SURMOF with a co-facial stacking (~5.8 Å) of the NDI(OEt)₂ cores, leading to an undesirable H-aggregate type arrangement (See method section and Supporting Figure 2).” Supporting Figure 2 shows a DFT-optimized trimer but neither MD-optimized geometry nor SURMOF.

Response: The referee is correct. In order to account for the criticism of the referee we have modified the text in the manuscript (highlighted in yellow in the revision). The revised text is as follow: “Using the DFT method we have optimized a trimer model of NDI(OEt)₂ (A). These

calculations yielded a co-facial stacking ($\sim 5.8 \text{ \AA}$) of the NDI(OEt)₂ cores, an undesirable H-aggregate type arrangement (See method section and Supporting Figure 2).”

A combination of molecular dynamics (MD) based simulation and density functional theory (DFT) predicts a NDI(OEt)₂ geometry in the Zn-(NDI(OEt)₂) (A) SURMOF with a co-facial stacking ($\sim 5.8 \text{ \AA}$) of the NDI(OEt)₂ cores, leading to an undesirable H-aggregate type arrangement (See method section and Supporting Figure 2).”

Line 200

Is it correct to say that Supporting Figure 5 illustrates the change in dihedral angle between R=H and R=iPr? If so, please replace ‘rotational angle’ by ‘dihedral angle’ in line 200, and reconsider the caption of Supporting Figure 5, for consistency with the words ‘dihedral angle’ and ‘rotation angle’ as defined in Supporting Table 1.

Response: We are grateful to the referee for drawing our attention to this point. First, here we provide proper definitions of the angles. We define the dihedral angle (α_1) as the angle between the cNDI core unit and the plane defined by the phenyl groups. There is a second dihedral angle (α_2) between the phenyl group and the carboxylate group. α_2 is basically zero for the isolated linker. If there were no interactions between adjacent linkers, the dihedral angle α_1 would be identical to the rotation angle θ (angle between cNDI plane and the carboxylate plane). Now, because of interactions between adjacent linkers, which are simulated by the GROMACS MD, α_2 starts to deviate from 0 (i.e. there is a torsion between carboxylate plane and phenyl plane, as shown in figure 5b in supporting information). As a result, for the MD optimized structure, the dihedral angle α_1 and the rotational angle θ can be different.

We have added a description on the dihedral angles (α_1 and α_2) to the method section.

We have also reworded the caption of figure 5 to make this point clear.

- Figure 1

Please keep the same colour code throughout.

Fig 1a : Ph are blue, zinc is green ; Fig 1b : Ph are green

Response: We have modified Fig. 1 – now the color scheme should be consistent.

Good.

Fig 1a : R groups are red ; Fig 2a : R groups are blue

Response: These problems have been removed.

Good.

- Figure 1b

* 'TDM-TDM coupling': please expand TDM acronym once (or in text line 140)

Response: We have added the expanded acronym in text line 140.

Good.

* caption 'Coulomb coupling vs rotation angle theta (as shown in Figure 1a at the right hand side)': theta is in fact shown in the inset of Figure 1b

Response: This problem has been removed.

Good.

* what does the green line correspond to ?

Response: The green line corresponds to the calculated TDM-TDM couplings for different rotation of NDI(OEt)₂, as mentioned in the figure caption.

If I understand correctly (after seeing the additional part of Supporting Figure 3), the line is the coupling, and you have added a few discrete marks to show examples for a few specific R groups.

Line 128 'Crystal structure prediction of NDI(OEt₂)-assembly'

Again if the calculations are not made on SURMOF structures, this title is misleading. Something like 'Structural prediction of a library of NDI assemblies' would be more correct, without diminishing the quality or the impact of the work.

Response: As mentioned before, we have used a MOF-constructor to assemble a SURMOF-2 from the ditopic linkers and then used MD-simulations to optimize the unit cell size along the [010] crystallographic direction. We agree that this is not a full structure determination. Following the suggestion of the reviewer, we have revised the line 128 as follows: "Structural prediction of a library of NDI(OEt)₂ assemblies".

Thank you for clarifying this point.

- Line 135 'co-facial stacking of the cores'

Please report the interchromophoric distance (in the text or on Supporting Figure 2)

Response: The missing information has been added to text.

Good.

- Line 141 'closely spaced chromophores'

Please report the interchromophoric distance

Response: The missing information has been added to text.

Good.

- Line 143 'placed at a distance (x) of 6.8 angstroms'
Is that what 'closely spaced' means ? This is significantly larger than the 5.4 angstroms distance of ref 39.

Where does this number come from ?

Response: This number has been obtained by determining the [010] unit cell parameter from the MD calculations for a non-rotated (R=H) cNDI linker. This value is larger than the 5.4 Å reported previously since the linkers are quite different.

Fair enough (Gromacs step ii). So why don't you write this first sentence somewhere, for clarity?

Response: We have followed the suggestion of the referee and have added a sentence in the computational method describing the origin of inter-linker distance.

Line 145 'for theta > 55.4°'

Where does this number come from? Isn't there a plot missing ? Second half of Supporting Figure 3 'angle dependence of monomers', Jmn vs theta ?

Response: This number comes from Figure 1b where the angle dependence of the Coulomb coupling for a fixed interchromophoric distance of 6.8 Å is displayed (green line). The same plot is now also introduced in the Supporting Figure 3.

Good.

- Line 158 'apparently more bulky groups yielded a smaller angle of theta'
In the methyl/benzyl comparison, this sentence should mention that they both induce similar dihedral angles (according to Supporting Table 1). Otherwise the comparison does not hold.

Response: This problem has been fixed.

• Does this correspond to the modifications in line 122 and lines 169-170 ? If so, then I disagree the problem is not fixed. Steric bulk induces changes both in dihedral and rotation (theta) angles. The comparison on rotation angles only does not hold without mentioning that the dihedral angles are similar.

Response: We agree with the referees comment. In accord with the suggestion, we have now mentioned the dihedral angles (α_1) in the related text (line 122 and 169-170), and highlighted it in yellow.

• Besides I would have thought that 34° and 41° were significantly different, whereas they are now quoted as 'similar'. Why?

Response: We agree with the referee. The wording has been corrected.

- Line 158 'methyl (9)'
In Figure 2a, methyl is 8

Response: This problem has been fixed.

Good.

Line 159 'benzyl (2)'
In Figure 2a, benzyl is 6

Response: This problem has been fixed.

Good.

- Line 172: "we chose three R groups, two from the borderline region in Figure 1
The borderline region is well defined on Figure 2b (blue square), but how is it defined in Figure 1?"

Response: Corrected accordingly.

- Line 167, figure 2b
I understand that the rotation angles were obtained from Gromacs-optimized trimers, is this correct? If so, this should be clarified.

Response: Yes, those were obtained from GROMACS. We have augmented the text of the figure caption to make this point clear.

Good but please make it completely clear, the caption should be able to stand alone without constantly requiring to jump to the computational details. It's only a matter of adding the word 'trimers'.

Response: We have followed the suggestion and made the corresponding changes to the figure legend.

- Line 195 'Supporting Information Figure 5' is in fact Supporting Figure 6

Response: The supporting figures have been revised accordingly.

Good.

- Line 204 'Supporting Information Figure 6' is in fact Supporting Figure 5

Response: The supporting figures have been revised accordingly.

Good.

Line 207 'vibronic overtones at 436 and 472 nm'

Does the vibrational spacing fit with a specific vibrational mode? Which one? Does it also fit with the simulated spectra shown in Supporting Figure 7?

Response: Yes, the vibronic features in the solvated cNDI-units can be related to vibrational modes, as has been discussed previously in the literature (DOI: 10.1039/c0cc00078g). However, this is not relevant for our work, since the vibronic features changed in the corresponding SURMOF data (as a result of strong excitonic coupling, see DOI:10.1002/anie.201708267). Since vibronic couplings were not included in the simulations, a direct comparison is not meaningful here. Please note also, that the absorption bands are shifted in comparison to experimental values, evidently due to the presence of solvation effects in the experimental results for the solvated monomers.

My suggestion was indeed to include vibronic coupling in the UV-vis simulation of monomer and dimers (Supporting Figure 7). This calculation could have produced a valuable addition.

Response: We have validated the predicted J-aggregation by experimental evidences. Simulating the vibronic couplings represents a major computational effort which clearly goes beyond the scope of the present work. Please note that the computational effort used just aims to screen the suitable "SCU" library. This has been a success, as evidenced by the good luminescence properties of the final SURMOF. The goal of this paper is not to provide a detailed theoretical analysis of the UV-Vis spectra of NDI compounds. We fully agree with the referee, that for a quantitative comparison between experiment and theory vibronic effects should be taken into account.

Line 212 'We noted that the coupling energy for A and 9A or 15A did not differ markedly (Figure 1b, ~50 meV)'

The data for R=iPr (as in SURMOF 15A) is shown on Figure 1b and is slightly negative (roughly -20meV) ; the data for R=Et (as in SURMOF 9A) is shown on Figure 1b and is slightly positive (roughly 20 meV) ; the data for R=H (as in SURMOF A) is not shown on Figure 1b. This doesn't make sense.

Response: The referee is correct. We have included the predicted θ for R=H by GROMAC-2018.4 in Figure 1b.

Thank you for adding the mark for R=H.

However I still don't understand why you are saying that the values are similar and close to 40 meV (line 219). To me they are neither similar, nor close to 40 meV.

The values I can read on Figure 1b are about +20 meV for R=Et, and about -20 meV for R=iPr. So please correct lines 218-219 accordingly, or explain again why your sentence is correct as it is. Those numbers can also be seen more easily on Figure 4d, which, by the way, has been replotted.

Response: In following the suggestion of the referee, we have now corrected the lines 218-219 and highlighted in yellow.

- previously plotted between -12 and +32 meV, with three data points estimated at -8; +18; +30;
 - now plotted between -20 and +25 meV, with three data points estimated at -16; +21; +23.
- Why and how did you replot this figure?

Response: We replotted the figure because there was a minor error (coupling values were not correctly mentioned) as noted in the previous version of the rebuttal letter. The figure was replotted with the TDM-TDM coupling values as shown in Figure 1b (obtained by MD simulation).

Line 216 'To explain this behavior we employed time dependent density functional theory (TD-DFT) to simulate the electronic absorption spectra of the NDI(OEt)₂ dimers with rotation angles 34 and 60° for H and J-type coupling, respectively (Supporting Information Figure 7 and 8)'

* How were these angles chosen ? The angles reported in Supporting Table 1 for R=Me or Et and R=iPr could have been chosen advantageously. Also see the vague expression 'certain angle' line 317.

Response: We are grateful for this comment by the referee. Briefly, we took two rotation angles (34 and 60 °), which clearly represent different aggregation types, H and J, respectively. A corresponding statement has been added to the manuscript.

Good.

* What was the interchromophoric distance in these TD-DFT calculations? Also see the vague expression 'proper distance' line 316.

Response: We have added the requested information (6.8 Å) to the manuscript.

Good.

* What is the origin of the additional red shoulder in the computed absorption spectrum of the 60°-tilted dimer ? If this shoulder is of vibronic origin, a vibrationally-resolved spectrum should be computed to confirm it. If, on the other hand, this shoulder is due to an additional electronic transition, it would be interesting to identify it.

Response: The origin of the additional transition at 485.62 nm in the absorption spectrum is an electronic transition. The known fact is that due to the parallel transition dipole orientations within coordinated chromophores in close spatial proximity low lying transitions are forbidden in the case of non-emissive H-aggregates. However, when two chromophores form a J-aggregate and two transition dipoles become almost collinear, this dark state becomes allowed and low lying visible absorption peaks are observed in the spectrum (**DOI: 10.1039/C0DT01226B**). From the comparison of the absorption spectra for the case of 34° and 60° in the Supporting Figure 8 (c and d), there is a visible rise in intensity of the previously dark singlet transition S1. From the analysis of the leading contributions for S1 transition it can be seen that it is dominated by HOMO-LUMO transition localized on central part of both chromophores.

Thank you for clarifying this point, I fully agree. The revised Supporting Figure 8 b) c) d) now allows the reader to compare the UV-vis absorption spectra for three different theta values, which is highly illustrative. Please add this explanation in the main text (lines 225-227) instead of the current very poor comment : “But, in J-type a new absorption shoulder was observed in the longer wavelength region (~485 nm).”

Response: Following the suggestion of the referee, we have added an additional sentence to the main text.

Line 223

Why are 8A and 9A so different in absorption (Figure 4a) but so similar in emission (Figure 4b,c) and structure (Supporting Table 1) ?

Response: 8A and 9A absorption spectra are different because their transition dipole moment (TDM) couplings are different (in line with the theoretical predictions).

I disagree with this. The computed theta values for R=Me and R=Et are very similar (40.9 vs 42.5°, Supporting Table 1) so the TDM-TDM couplings should be very similar too.

Response: Yes, we agree with the reviewer that the TDM-TDM coupling values for R= -Me and -Et are very similar (+23 and 21 meV).

One would thus expect that the experimental values are also similar. The reason why the experimental absorption spectra are quite different is not clear to us. To understand the difference, thorough theoretical analyses of the spectra and structure are necessary, preferentially

including vibrational coupling etc. However, such calculations are beyond the scope of the present work. We would like to repeat that we realized only a rather simple calculation scheme for screening our library of SCUs.

Although the steady state emission spectra (Figure 4b) of 8A and 9A do not differ significantly, there is an evident difference in PLQY (Figure 4d) for these two species. A more detailed answers to this questions would require a thorough theoretical analysis, which we feel is beyond the scope of the present paper. In fact, the message of the paper is that we use computational screening to find side groups which turn the “dark” SURMOF into an emissive one. The success of this approach is evident from the experimental observation that the SURMOF fabricated from the optimized linkers indeed is highly emissive.

OK.

- Figure 4a

The colour code seems wrong with an inversion between 8A and 9A (inconsistency with the text and with the caption of Figure 3)

Response: We are grateful to the referee for pointing out this problem. Figure 3 has been corrected and the color codes are now consistent.

Good.

- Figure 4b

The partly hidden caption should be completely hidden

Response: Nothing is hidden in Figure 4b, it is a relevant legend.

My mistake, sorry. I got confused between 15 and 15A.

Figure 4d

* Please provide the emission spectra corresponding to Figure 4d-inset(ii) as Supporting Information

Response: We have followed the suggestion of the referee and have added the requested material to the supporting information.

• Do you mean the new Supporting Figure 10, that was previously shown as an inset of Figure 4d? I suggested you to show the emission spectra, in addition to the photos. The PL spectrum of 15 is indeed shown on Figure 4b, but the PL spectrum of the linker with R=H is not shown.

Response: Now we have added the PL spectra (Figure 11) of the linkers in supporting information.

• Besides, the caption of Supporting Figure 10 is incomplete with respect to the caption of the previous Figure 4d : “SURMOF thin films deposited on a quartz substrate” has become “SURMOFs”.

Response: We have followed the suggestion of the referee and have made the corresponding changes to the manuscript.

* are the TDM-TDM coupling values shown on Figure 4d supposed to match some of those shown on Figure 1b ? It is not clear to me what the difference between these two sets of data is.

Response: We thank the reviewer for pointing out this issue. We have chosen the TDM-TDM coupling values from figure 1b, which were computed for the initially predicted rotation angles of **8A**, **9A** and **15A**.

Do you mean that the initial Figure 4d used other TDM-TDM values because they used other rotation angles? Angles from MD-optimized trimers, instead of angles from DFT-optimized trimers? Or yet other angles?

Response: The TDM-TDM values used in figure 4d is from figure 1b (values obtained by MD simulations). We have mentioned this in the figure 4d caption and highlighted in yellow.

Figure 4d shows a correlation of prediction and experimental results. In the revised figure 4d, the values can be correlated perfectly with figure 1b and supporting table 1. We have also mentioned in the figure caption of figure 4d, that the TDM-TDM couplings plotted are from Figure 1b.

OK.

Line 264 'crystal structure prediction'

Same remark as above : at this stage I can't say if and how any SURMOF structure was computed at all.

Response: As explained above we feel that using this term is justified. We have used approximate methods to predict SURMOF structures, we did not claim that the SURMOF structure was determined using a full DFT structure optimization.

My concern is that you are showing monomers, dimers, trimers, but you are not providing any SURMOF structure, unlike DOI: 10.1038/srep00921.

• Figure 1b, Figure 2b, Figure 4d, Supporting Table 1, Supporting Figure 2, Supporting Table 2, Supporting Figure 5 : concern trimers

• Supporting Figure 3, Supporting Figure 7, Supporting Figure 8, Supporting Table 3 : concern dimers

I'm not asking at all for a full DFT structure optimization. I'm only asking for clarity, consistency and rigour.

Response: We have followed the suggestion and changed the text (highlighted in yellow) to make it clear that no full DFT structure optimizations were carried out for the SURMOFs. Instead, we used a more simple crystal structure prediction (CSP) algorithm. A proper definition of our CSP approach has been added to the main text (see above). We feel that this modification should fully account for the concerns of the referee.

Line 267 '+40 meV'

* Was this computed for a dimer with R=H or for SURMOF A?

Response: This was computed for a dimer with R=H.

OK so please specify it, because at the point where this is written, it seems you are talking about data for the SURMOFs themselves.

- In addition, why did you change the +40 meV value into +22 meV?
- And how do you get 'almost a factor of two' between 22 and 16?

Response: We have followed the suggestion of the referee, the sentence has been modified accordingly.

The previous value of +40 meV was not correct, we have corrected the value. We are grateful to the referee for pointing out this problem.

In following the suggestion of the referee, we have removed the sentence "almost a factor of two".

* Similarly, '-15 meV', was this computed for a dimer with R=iPr or for SURMOF 15A?

Response: This was computed for a dimer with R=iPr.

Same remark as above, please specify it.

Response: We have followed the suggestion of the referee, the sentence has been modified accordingly.

Line 293 'Optimization of the layer distance via calculating the potential energy of neighboring linkers as a function of layer spacing (5.0~8.0 angstroms)'

Is this performed on dimers or on MOF structures? Since step (i) concerns monomers and step (iii) concerns trimers, I would rather think that step (ii) concerns dimers, in which case there are no calculations on SURMOF structures.

Response: This optimization was performed on a dimer of cNDI with related SCU attached to it.

Thank you for clarifying this point.

- Line 307 'Taking the optimized potential energy value as criteria, the geometry obtained via SA-MD is substantially better than without SA-MD optimization'

Please give elements of comparison (distances, energies).

Response: Following is a comparison of potential energy values obtained with and without SA-MD: By Using SA-MD optimization, for linker **13**, the monomer (step i) potential energy value decreased by 32 kJ/mol, and for trimer (step iii) it is 53 kJ/mol.

OK.

- Line 316 'proper distance'

For clarity, please recall these numbers for each R group.

Response: The requested information has been added to the manuscript.

Thank you for clarifying this point.

- Line 317 'certain angle'

For clarity, please recall these numbers for each R group.

Response: The requested information has been added to the manuscript.

Thank you for clarifying this point.

- Line 326 'Becke D3 dispersion'

should be 'Grimme D3 dispersion'.

Response: We have changed the wording in accord with this suggestion.

Good.

- Line 327 'high quality grid'

Please be more specific. Also please specify which program was used for the DFT and TD-DFT calculations.

Response: We have added the requested information to the paper.

Good.

- Line 391 'photochemical' should be 'photomechanical'.

Response: Has been corrected.

Good.

- Supporting Information Table 1

If I understood correctly, these calculations were performed on Gromacs-optimized trimers. For clarity this information could be recalled in the Table caption.

Response: We have added the requested information to the manuscript.

Good but please make it clear as well in the Supporting Table 1 caption, which should be able to stand alone. It's only a matter of adding the word 'trimers'. In fact the rotation angles in Supp. Table 1 are the ones that are plotted on Figure 2b, so please indicate this link in the caption of Supp. Table 1 too, in order to help the reader.

Response: We have followed the suggestion of the referee and have done the requested changes.

Supporting Information Figure 2

* Please specify interchromophoric distance

* If the calculation indeed concerns a SURMOF structure, how was it performed and where are the zinc ions ?

Response: The values for the inter-chromophore distance have been added to the manuscript. The calculations were performed for a trimer model, without considering the zinc paddle-wheel units. This information has been added to the legend of the figure.

The caption now seems correct (the initial one wasn't), and consistent with the computational methods.

Supporting Information Figure 3

The angle dependence of the Coulomb coupling should be shown.

Response: Please note that the angle dependence is already present Figure 1b of the original manuscript. In order to account for the criticism of the referee, we have added this information to Figure 3 (bottom).

My question was related to the fact that Supp. Figure 3 seemed incomplete as it indicated "angle dependence of monomers" without any graphical support (cf red oval that I added on my screen capture).

Response: We have followed the suggestion of the referee.

Supporting Information line 151 'The shift in (010) diffraction from A to the other SURMOFs indicate a slight changes (~ 0.1 angstrom) in the inter-sheet distances'
Computed intermolecular distances given in Supporting Table 1 (presumably in trimers, not in SURMOFs) show that the intermolecular distances vary by 1 angstrom (from 5.8 angstroms for R=H to 6.8 angstroms for R=iPr). Isn't this a concern to you that this doesn't fit with the experimental intermolecular distances, which are the same for all SURMOFs?

Response: We are grateful to the referee for pointing out this problem. The sentence has been revised. From A to 15A, the change in inter-sheet distance is $\sim 1 \text{ \AA}$, but not $\sim 0.1 \text{ \AA}$ (also mentioned in the main text). So the calculation and experimental values agree well.

Good to know!

- Supporting Information line 155

For clarity please specify 'distances as in respective experimental SURMOF structures'.

Response: We have modified the manuscript in accord with this suggestion.

Please recall these distances in Supporting Table 2 as well (e.g. additional column). It's really irritating to have to chase numbers all the time!

Response: We have followed the suggestion of the referee. The requested information has been added to the manuscript.

Supporting Information Table 2

How do you explain that the angles in Supp. Table 1 (from Gromacs-optimized trimers) are so different from the angles in Supp. Table 2 (from DFT-optimized trimers)? $24\text{-}28^\circ$ instead of $41\text{-}42^\circ$, and 53° instead of 63° .

Given the angular variation of the TDM-TDM coupling, it is crucial to have reliable angles, especially in the region of the magic angle.

Please rephrase lines 197-198 accordingly: "The DFT simulated linker orientations in all four structures predict different rotation angles (Supporting Information Table 2)."

Response: The optimizations results of MD (which is based on a force-field) and DFT are expected to be different. The DFT values should, of course, be more accurate. We do not know the precise reason why these differences are so large.

In our CSP approach we use MD for a quick screening and then did the further (computationally more expensive) optimizations by DFT only for the experimentally synthesized structures. Hence, this difference does not affect our prediction strategy. We would like to repeat that our CSP approach does not aim for a thorough quantitative analysis of absorption spectra of NDI structures. The CSP approach is used to identify promising structures.

Line 197-198 have been rephrased and now read: “The DFT simulated linker orientation angles are different (as expected) from the GROMACS-optimized values, see Supporting Information Table 2.

Supporting Information Figure 6

* The caption says 'DFT simulated geometries of A and 15A', which is very confusing with respect to the Computational methods section. Is there any zinc in these calculations? It seems as if these calculations were performed on monomers.

Response: We have modified the manuscript in accord with this suggestion.

The caption now seems correct (the initial one wasn't), and consistent with the computational methods.

* According to Supporting Table 1, the dihedral angle between the NDI core and the Ph substituents is 69.1° for R=H and 87.4° for R=iPr. This is not obvious from Supporting Figure 6.

Response: We have modified the manuscript in accord with this suggestion.

Good.

- Supporting Information line 172

* Please specify interchromophoric distance in the dimers.

Response: We have modified the manuscript in accord with this suggestion. In the revised version these distances are mentioned.

OK

* Were they optimized as in Gromacs step (ii) ?

Response: Yes, we obtained this from GROMACS-2018.4 based calculations (in step ii).

Good.

- Supporting Information Figure 8

* Please specify interchromophoric distance in the dimer.

Response: Now the distances are mentioned.

Good.

* Please show 60° angle on the figure. Is this the theta angle of Figure 1b?

Response: Now we have added this in the revised version.

Good.

Unfortunately, we do not understand the question “Is this the theta angle of Figure 1b?”. Figure 1b and SI Figure 8 are clearly described in the corresponding figure captions.

OK

* Is the image at the left of the computed absorption spectrum a sort of top view of the one shown in the inset of the spectrum?

Response: Yes, it is the top view. It is described in the revised version.

Good.

* Please show the full spectrum (the red region is currently cut)

Response: The red parts of the spectra were deliberately omitted due to the fact that there are no electronic transitions. The additional Table 3 in the Supporting Information is now added, with the overview of the all calculated transitions and corresponding wavelengths. This clearly shows that lowest singlet transitions for all chromophore assemblies lay below 500 nm.

I agree that there are no additional peaks above 500 nm. However sometimes tails or shoulders (or absence of) are important to show.

* 'top' and 'bottom' do not correspond to the current display

Response: We have modified the manuscript in accord with this suggestion.

Good.